# QLLM: Accurate and Efficient Low-Bitwidth Quantization for Large Language Models

**Jing Liu**[1,2][*] **Ruihao Gong**[2,3]**, Xiuying Wei**[2,4]**, Zhiwei Dong**[2,5]**, Jianfei Cai**[1]**, Bohan Zhuang**[1][†]

[1]ZIP Lab, Monash University    [2]SenseTime Research    [3]Beihang University
[4]School of Computer and Communication Sciences, EPFL
[5]University of Science and Technology Beijing

## Abstract

Large Language Models (LLMs) have demonstrated unparalleled efficacy in natural language processing. However, their high computational demands and memory overheads hinder their broad deployment. To address this, two quantization strategies emerge, including Quantization-Aware Training (QAT) and Post-Training Quantization (PTQ). For LLMs, the billions of parameters make the QAT impractical due to the prohibitive training cost and thus PTQ becomes more prevalent. In existing studies, activation outliers in particular channels are identified as the biggest challenge to PTQ accuracy. They propose to transform the magnitudes from activations to weights, which however offers limited alleviation or suffers from unstable gradients, resulting in a severe performance drop at low-bitwidth. In this paper, we propose QLLM, an accurate and efficient low-bitwidth PTQ method designed for LLMs. QLLM introduces an adaptive channel reassembly technique that reallocates the magnitude of outliers to other channels, thereby mitigating their impact on the quantization range. This is achieved by channel disassembly and channel assembly, which first breaks down the outlier channels into several sub-channels to ensure a more balanced distribution of activation magnitudes. Then similar channels are merged to maintain the original channel number for efficiency. Additionally, an adaptive strategy is designed to autonomously determine the optimal number of sub-channels for channel disassembly. To further compensate for the performance loss caused by quantization, we propose an efficient tuning method that only learns a small number of low-rank weights while freezing the pre-trained quantized model. After training, these low-rank parameters can be fused into the frozen weights without affecting inference. Extensive experiments on LLaMA-1 and LLaMA-2 show that QLLM is able to obtain accurate quantized models efficiently. For example, QLLM quantizes the 4-bit LLaMA-2-70B within 10 hours on a single A100-80G GPU, outperforming the previous state-of-the-art method by 7.89% on the average accuracy across five zero-shot tasks. Code is available at ZIP Lab and ModelTC.

## 1 Introduction

Recently, Large Language Models (LLMs) such as GPT-4 (OpenAI, 2023) and LLaMA (Touvron et al., 2023a;b) have achieved unprecedented advancements in natural language processing (NLP). These models excel in a range of tasks, from advanced reasoning in code and mathematics to classification and question answering. However, their extraordinary performance is accompanied by substantial computational demands and vast model sizes. For example, GPT-3 (Brown et al., 2020), the precursor to GPT-4, already contains a stunning 175 billion parameters, requiring a minimum of 325 GB of memory for storage in half-precision (FP16) format. This necessitates the use of at least 5×80GB NVIDIA A100 or 8×48GB NVIDIA A40 GPUs during the inference phase. As a result, deploying these models to real-world applications poses significant challenges.

---

[*]Work done during an internship at SenseTime Research.
[†]Corresponding author. Email: `bohan.zhuang@gmail.com`

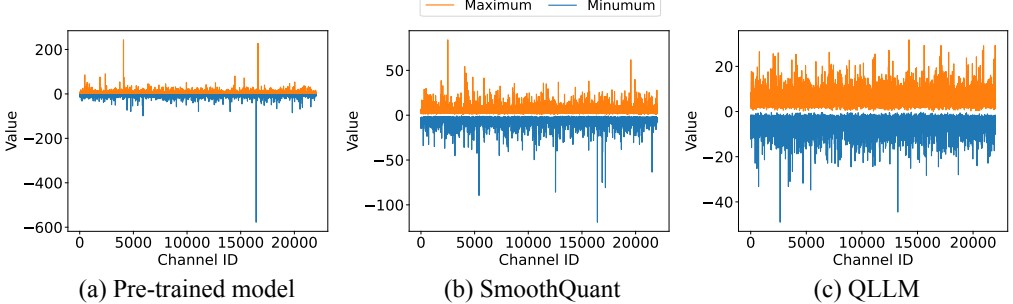

Figure 1: An illustration of the channel-wise maximum and minimum values for the input activations of a linear layer in LLaMA-65B for (a) original pre-trained model (b) after SmoothQuant (Xiao et al., 2023) and (c) after our channel reassembly.

In light of the aforementioned challenges, network quantization (Zhou et al., 2016) emerges as a compelling solution, which maps weights and/or activations to lower-bit representations, resulting in a much lower memory footprint and faster inference. Existing quantization methods for LLMs can be classified into two types: quantization-aware training (QAT) (Liu et al., 2023) and post-training quantization (PTQ) (Wei et al., 2022b; 2023; Xiao et al., 2023). Although with promising performance, QAT suffers from unbearable training costs as it needs to fine-tune the whole quantized model with quantization parameters using a large amount of data, rendering it impractical for the efficient deployment of LLMs. This practical limitation has shifted the spotlight towards PTQ which only uses a little data to tune the quantized weights. However, when it comes to extremely low-bitwidth quantization for LLMs, *e.g.*, 4-bit weight and/or activation quantization, existing PTQ methods (Xiao et al., 2023; Dettmers et al., 2022) suffer from significant performance degradation.

Recent studies (Dettmers et al., 2022; Xiao et al., 2023; Wei et al., 2023) have revealed a unique pattern in LLMs' activations that is they contain specific outlier channels with significantly large magnitudes. This renders existing quantization methods less effective, as the outliers amplify the quantization range of layer activations, causing the vast majority of normal activation values to be quantized imprecisely and consequently leading to notable performance degradation. This issue will worsen with the prevalent use of layer-wise or token-wise activation quantization, a common practice for maximizing hardware efficiency. To tackle this challenge, recent studies (Xiao et al., 2023; Wei et al., 2022b; 2023; Shao et al., 2023) have focused on smoothing activation outliers by transitioning the magnitudes from activations to weights through a mathematically equivalent transformation. Such a transformation can be learned using either gradient-free methods (Xiao et al., 2023; Wei et al., 2022b; 2023) or gradient-based methods (Shao et al., 2023). However, as shown in Figure 1, for exceedingly pronounced activation outliers (those 50 × larger than others), the former offers only limited alleviation while the latter suffers from unstable gradients. As a result, both methods leads to significant performance degradation in low-bitwidth quantization. To compensate for the performance drop of quantization, a widely adopted PTQ strategy (Wei et al., 2023; Shao et al., 2023; Yao et al., 2022) further proposes to tune the quantized LLM directly by minimizing the block-wise reconstruction error. In LLMs, the tuned block refers to the Attention-FFN module. However, considering the huge number of parameters in an LLM, this approach still requires substantial training overheads and demands a significant amount of GPU memory.

In this paper, we propose QLLM, an accurate and efficient low-bitwidth post-training quantization method tailored for LLMs. To handle the outlier issue, we introduce a gradient-free channel reassembly technique that redistributes the large activation magnitude of the outlier channels across the channels. Specifically, we first disassemble the outlier channels into several sub-channels. By spreading the magnitude of outliers, it ensures a more uniform activation range across channels, facilitating a balanced and precise quantization and thus improving the performance of quantized LLMs. We then introduce channel assembly, which fuses similar channels together to maintain the original channel count. Moreover, given the varying outlier patterns across different layers and the existence of extreme outliers, we propose an adaptive strategy to determine the optimal number of disassembled channels for each layer, which is based on minimizing the reassembly error between the original output activations and the counterpart with the reassembled input activations.

To further improve the performance of the quantized LLMs, motivated by low-rank parameter-efficient fine-tuning paradigm LoRA (Hu et al., 2022; Dettmers et al., 2023a), we further propose

an efficient gradient-based error correction strategy that freezes the pre-trained model and introduces a small set of learnable low-rank weights into each layer of the LLM. Then, QLLM learns the low-rank weights by minimizing block-wise quantization error sequentially. Owing to the reduced number of trainable parameters, both the training time and GPU memory requirements are significantly reduced. Such efficiency gain enables us to perform a multi-block reconstruction that simultaneously reconstructs a collection of consecutive Attention-FFN blocks, further mitigating the quantization error accumulation during propagation in low-bit LLMs. Notably, after training, these learnable low-rank weights can be seamlessly merged with the frozen weights followed by quantization, thereby ensuring no additional computational burden during inference.

Our contributions can be summarized as follows: 1) We introduce a simple yet effective channel reassembly method to suppress activation outliers in LLMs, which is accomplished by initially disassembling the outlier channels to make activations more quantization-friendly and subsequently merging similar channels so as to preserve the original channel count for efficiency. We also propose to determine the optimal number of disassembled channels for each layer, considering the diverse outlier patterns across layers and the presence of extreme outliers. The overall process is gradient-free and enjoys high efficiency. 2) An efficient error correction mechanism is proposed to further enhance the gradient-free channel reassembly. It leverages the learning of low-rank parameters to counteract quantization error in a structured way, leading to a substantial reduction in training time and GPU memory requirements without incurring any additional inference overhead. 3) Extensive experiments show the promising performance and training efficiency of QLLM. For example, QLLM quantizes 4-bit LLaMA-2-70B within 10 hours, and outperforms previous SOTA methods by 7.89% on the average accuracy across five zero-shot tasks.

## 2 RELATED WORK

**Network quantization.** Network quantization (Zhou et al., 2016) which represents the weights, activations, and even gradients with low precision, is an effective method to reduce the model size and computational burden. Existing techniques fall into two primary categories: quantization-aware training (QAT) (Esser et al., 2020; Kim et al., 2021; Li et al., 2022) and post-training quantization (PTQ) (Nagel et al., 2020; Li et al., 2021; Wei et al., 2022a). QAT incorporates the quantization process directly into the training phase and jointly learning the quantizer as well as model parameters (Zhang et al., 2018; Jung et al., 2019; Choi et al., 2019; Bhalgat et al., 2020; Esser et al., 2020; Liu et al., 2022) with the help of straight-through estimator (STE) (Bengio et al., 2013), which greatly mitigates the accuracy degradation caused by compression. However, the training cost of QAT can be prohibitively high, primarily because it requires fine-tuning the quantized model on the original training dataset of the pre-trained model. PTQ offers a less resource-intensive alternative, allowing models to be quantized after being fully trained with only a small amount of data. To reduce the performance drop, several methods have been proposed to perform layer-wise (Nagel et al., 2019; 2020; Wu et al., 2020; Hubara et al., 2020; Li et al., 2021) or even block-wise calibration (Li et al., 2021). Further innovations delve into outlier mitigation, adopting strategies like clipping (Banner et al., 2019; McKinstry et al., 2019; Choukroun et al., 2019) or value splitting (Zhao et al., 2019) for weights and activations to improve the precision by allocating more bits to the intermediate values. However, for LLMs, a recent study (Liu et al., 2023) has found that MinMax quantization, which maintains the full value range, performs better than clipping-based methods, as outliers are critical to the performance. Different from these methods, our QLLM targets quantization for LLMs.

**Quantization on LLMs.** Given constraints such as limited training data and intensive computational demands, prevailing quantization techniques for LLMs are primarily based on PTQ. Existing LLM quantization approaches can be classified into two categories: weight-only quantization (Frantar et al., 2022; Park et al., 2023; Lin et al., 2023; Dettmers et al., 2023b; Chai et al., 2023; Cheng et al., 2023; Dettmers et al., 2023a; Kim et al., 2023; Chee et al., 2023; Lee et al., 2023) and weight-activation quantization (Dettmers et al., 2022; Xiao et al., 2023; Wei et al., 2022b; 2023; Yao et al., 2022; 2023; Yuan et al., 2023; Liu et al., 2023; Wu et al., 2023). The former focuses on compressing the vast number of weights in LLMs to reduce the memory footprint, while the latter compresses both weights and activations into low-bit values, aiming to accelerate computation-intensive matrix multiplication. To handle the different value ranges of weight matrices, recent studies have delved into more fine-grained quantization, such as channel-wise quantization (Frantar et al., 2022) or group-wise quantization (Frantar et al., 2022; Lin et al., 2023). To further compensate for the performance drop for extremely low-bitwidth quantization, QLoRA (Dettmers et al., 2023a), and INT2.1 (Chai et al., 2023) introduce additional full-precision weights (Yao et al., 2023). While our

method also presents a small set of low-rank weights, it stands apart from QLoRA and INT2.1 as our learnable parameters can be reparameterized into pretrained weights followed by quantization. Recent research (Dettmers et al., 2022) has shown that activation outliers exist in some feature dimensions across different tokens. Several works (Wei et al., 2022b; 2023; Xiao et al., 2023; Shao et al., 2023) have been proposed to migrate the quantization difficulty from activations to weights within the same channel, based on gradient-free methods (Wei et al., 2022b; Xiao et al., 2023; Wei et al., 2023) or gradient-based methods (Shao et al., 2023). However, when dealing with very pronounced activation outliers, the existing methods often show limited improvement or incur unstable gradients. In a notable difference, our proposed QLLMs method efficiently redistributes the large activation magnitudes of outlier channels among all channels, offering a distinctive approach compared to these existing methods.

## 3 PRELIMINARIES

**Basic notations.** In this paper, matrix is marked as $\mathbf{X}$ and vector is denoted by $\mathbf{x}$. The LLMs usually have two core parts: multi-head self-attention (MSA) layers and feed-forward network (FFN) layers, which are mainly composed of linear layers. Here, we give the formulation of linear layers at the output channel $k$:

$$\mathbf{y}_k = \sum_{i=1}^{M} \mathbf{x}_i \mathbf{W}_{ik}, \tag{1}$$

where $\mathbf{x} \in \mathbb{R}^M$ refers to input, $\mathbf{W} \in \mathbb{R}^{M \times N}$ denotes the weight, and $\mathbf{y} \in \mathbb{R}^N$ stands for the output. In this way, the numbers of input and output channels are $M$ and $N$, respectively.

**Quantization.** We adopt uniform quantization for both weights and activations because of its hardware-friendly nature (Jacob et al., 2018). For matrix $\mathbf{X}$ with floating-point values such as FP16 or FP32, the $b$-bit quantization quantizes it in the following way:

$$\mathbf{X}_q = \text{quant}(\mathbf{X}) = \text{clamp}\left(\lfloor \tfrac{\mathbf{X}}{\alpha} \rceil + \beta, 0, 2^b - 1\right), \text{where } \alpha = \tfrac{\max(\mathbf{X}) - \min(\mathbf{X})}{2^b - 1}, \beta = -\left\lfloor \tfrac{\min(\mathbf{X})}{\alpha} \right\rceil, \tag{2}$$

where the function $\text{clamp}(v, v_{\min}, v_{\max})$ clips any value $v$ into the range of $[v_{\min}, v_{\max}]$ and $\lfloor \cdot \rceil$ is a rounding operator that returns the nearest integer of a given value. Here, $\alpha$ denotes the scaling factor and $\beta$ represents the zero-point value.

Recent studies (Dettmers et al., 2022; Xiao et al., 2023; Wei et al., 2022b) point out that there are extremely large outliers in certain channels of activations in LLMs, which makes the quantization challenging to balance the accurate representation for large values and small numbers. To tackle this problem, some approaches (Bondarenko et al., 2021; Yuan et al., 2023) adopt fine-grained quantization scheme, which assigns different quantization parameters for different channels. However, such a way needs delicate kernel design and clearly increases computation overhead for inference. Also, some works Wei et al. (2022b); Xiao et al. (2023) propose to use channel-wise scaling between activation and weights, which still remains outliers under extreme cases, as shown in Figure 1.

## 4 PROPOSED METHOD

In this section, we propose the adaptive channel reassembly framework to redistribute input activation outliers across multiple channels. The framework consists of three components: channel disassembly for decomposing the outlier channel, channel assembly for balancing the efficiency, and an adaptive strategy to find the suitable reassembly ratio for each layer. The channel reassembly technique is gradient-free and efficient to implement. What's more, it can be equipped with a gradient-based and well-designed error correction module for further enhancement.

### 4.1 ADAPTIVE CHANNEL REASSEMBLY

#### 4.1.1 CHANNEL DISASSEMBLY

In this part, we introduce our channel disassembly to decompose the input outlier channels into several sub-channels, which can reduce the outlier magnitude and make the activations more quantization-friendly without altering the layer output.

Considering that outliers tend to be concentrated in specific channels across various inputs and the desire to preserve their information during quantization, we propose to break down these outlier channels into several sub-channels to redistribute their large values. Without loss of generality, by assuming the $M$-th channel as the outlier channel, we can disassemble it into $\frac{\mathbf{x}_M}{T}$ and replicate this

channel $T$ times, reducing the outlier magnitude by a factor of $T$. Simultaneously, it is also natural to duplicate the corresponding weight channel $T$ times, enabling us to maintain the equivalent output:

$$\mathbf{y}_k = \sum_{i=1}^{M-1} \mathbf{x}_i \mathbf{W}_{ik} + \underbrace{\frac{\mathbf{x}_M}{T} \mathbf{W}_{Mk} + \cdots + \frac{\mathbf{x}_M}{T} \mathbf{W}_{Mk}}_{T \text{ times}}. \tag{3}$$

The equation above produces the same output with the original linear layer equation in Eq. (1) and introduces an additional $T - 1$ channels for both the input and the weight.

Taking into account that the quantization range impacts accuracy, we introduce an outlier threshold, denoted as $\theta$, to identify the outlier channels and determine the number of sub-channels together, with $T = \lceil \max(|\mathbf{x}_M|)/\theta \rceil$. This approach ensures that channels with values smaller than $\theta$ remain unchanged with $T = 1$, while the magnitude of outliers are divided by $T$.

Our channel disassembly method allows us to retain outlier information with an equivalent output and ease the quantization difficulty with a much smaller value range. Its only drawback is the increase in the number of channels, which may lead to additional computational costs and will be addressed in the next subsection.

### 4.1.2 Channel Assembly

Note that the input channel count increases to $M + T - 1$ after channel disassembly. Given the substantial quantity of channels in LLMs, it is possible to omit some unimportant channels or merge similar input channels to keep the original channel count $M$ for efficiency while maintaining outputs. To achieve this, a straightforward method is to use channel pruning (Ma et al., 2023; Sun et al., 2023) that removes the unimportant channels directly. However, such method may result in substantial information loss, especially when $T$ is large. Motivated by recent studies (Bolya et al., 2023; Bolya & Hoffman, 2023) that combine similar tokens, we propose a channel assembly method that delves into merging $T - 1$ similar input channels. Given channels $i$ and $j$, in alignment with token merging techniques (Bolya et al., 2023; Bolya & Hoffman, 2023), our goal is to aggregate them by calculating the average of their input features, denoted as $\frac{\mathbf{x}_i + \mathbf{x}_j}{2}$ and utilizing the aggregated feature in subsequent computations, which is defined as:

$$\mathbf{x}_i \mathbf{W}_{ik} + \mathbf{x}_j \mathbf{W}_{jk} \approx \frac{\mathbf{x}_i + \mathbf{x}_j}{2} \left( \mathbf{W}_{ik} + \mathbf{W}_{jk} \right), \tag{4}$$

where $\mathbf{W}_{ik} + \mathbf{W}_{jk}$ represents the merged weight. With the aim of minimizing the information loss of channel assembly in Eq. (4), we can define a distance metric $D(i, j)$ between channels $i$ and $j$ as

$$D(i, j) = \left\| \frac{\mathbf{x}_i (\mathbf{W}_{ik} - \mathbf{W}_{jk})}{2} + \frac{\mathbf{x}_j (\mathbf{W}_{jk} - \mathbf{W}_{ik})}{2} \right\|_2^2, \tag{5}$$

where $\|\cdot\|_2$ represents the $\ell_2$ norm. The above distance metric takes into account the difference in both input activations and weights between the two channels.

With the channel distance defined, the next step is to determine which channels to aggregate efficiently, with the goal of reducing the total channel count by $T - 1$. To address this, we propose using bipartite soft matching (Bolya et al., 2023; Bolya & Hoffman, 2023) that first partitions the channels into two sets, each containing roughly equal sizes, and subsequently finds the $T - 1$ most similar pairs between these two sets (see Appendix A for details). Note that we do not assemble the channels that are disassembled from the outlier channels since they play a critical role in the performance of LLMs. After the channel reassembly, including both disassembly and assembly, we acquire the reassembled input activations that are more amenable to quantization, along with the corresponding reassembled weights for layer $l$.

### 4.1.3 Adaptive Reassembly

In this section, we present a method to adaptively determine the appropriate reassembly ratio for each layer. For channel disassembly, selecting a high value for $T$ with a small $\theta$ substantially reduces outlier magnitudes and benefits quantization, while resulting in a larger increase in channel merging error due to a higher merging ratio. Conversely, choosing a small $T$ with a large $\theta$ will not increase the channel count much, making it easier for the assembly stage to keep the information while likely still retaining outliers, causing significant quantization errors. Therefore, it is crucial to carefully determine the outlier threshold $\theta$ or the reassembly channel number $T$.

However, it is hard to choose $\theta$ in practice as distinct layers have different patterns of outliers, as shown in Figure D. Motivated by (Wei et al., 2023), we propose an adaptive strategy to find the optimal $\theta$ by minimizing the reassembly error between the original output activations and their counterparts generated with the reassembled input activations for each layer.

Note that our channel reassembly technique can yield the reassembled activation $\hat{\mathbf{X}} \in \mathbb{R}^{L \times M}$ with a sequence length of $L$, which can then be fed into a MSA layer or a FFN layer. For example, let us consider a case where $\hat{\mathbf{X}}$ is fed into a MSA layer. A standard MSA layer calculates queries, keys and values with three learnable projection matrices $\mathbf{W}_Q, \mathbf{W}_K, \mathbf{W}_V \in \mathbb{R}^{M \times N}$ as $\mathbf{Q} = \mathbf{X}\mathbf{W}_Q, \mathbf{K} = \mathbf{X}\mathbf{W}_K, \mathbf{V} = \mathbf{X}\mathbf{W}_V$, where $\mathbf{X} \in \mathbb{R}^{L \times M}$ represents the original input activation. Let $\hat{\mathbf{W}}_Q$, $\hat{\mathbf{W}}_K$, $\hat{\mathbf{W}}_V$ be the reassembled projection weights. In this way, the reconstructed queries, keys, and values can be formulated as $\tilde{\mathbf{Q}} = \text{quant}(\hat{\mathbf{X}})\text{quant}(\hat{\mathbf{W}}_Q), \tilde{\mathbf{K}} = \text{quant}(\hat{\mathbf{X}})\text{quant}(\hat{\mathbf{W}}_K), \tilde{\mathbf{V}} = \text{quant}(\hat{\mathbf{X}})\text{quant}(\hat{\mathbf{W}}_V)$. We then find $\theta$ by solving the problem as

$$\arg\min_\theta \left\| \text{Softmax}(\mathbf{Q}\mathbf{K}^\top)\mathbf{V} - \text{Softmax}(\tilde{\mathbf{Q}}\tilde{\mathbf{K}}^\top)\hat{\mathbf{V}} \right\|_F^2, \tag{6}$$

where $\|\cdot\|_F$ denotes the Frobenius norm. To solve problem (6) efficiently, we use grid search following (Choukroun et al., 2019; Wei et al., 2023) (see Algorithm 1 in Appendix for details).

## 4.2 Efficient Gradient-based Error Correction

Based on the above gradient-free adaptive channel reassembly, an efficient gradient-based error correction technique is further proposed for improving the performance of the quantized LLMs using a small set of calibration data.

Inspired by recent developments in parameter-efficient fine-tuning methods (Hu et al., 2022; Dettmers et al., 2023a), the efficient error correction introduces two low-rank parameters $\mathbf{A} \in \mathbb{R}^{M \times r}$ and $\mathbf{B} \in \mathbb{R}^{r \times N}$ with a rank of $r$ into each projection layer of our QLLM. Then, we can obtain the output $\mathbf{Y}$ of a quantized linear layer by $\mathbf{Y} = \text{quant}(\mathbf{X})\text{quant}(\mathbf{W}) + \text{quant}(\mathbf{X})\mathbf{A}\mathbf{B}$. Instead of directly tuning the quantized weights, we learn the introduced low-rank parameters by minimizing the reconstruction error between the original and the quantized outputs of the Attention-FFN block. Thanks to the reduced number of trainable parameters, both the optimization cost and GPU memory usage can be significantly reduced, Such efficiency gain allows us to further suppress the accumulation of quantization error during forward propagation via a structured reconstruction, *i.e.*, performing multi-block reconstruction for QLLM, which simultaneously adjusts a collection of consecutive Attention-FFN blocks by focusing on reconstructing the final block output.

After the reconstruction, we only need to store the quantized weight $\text{quant}(\mathbf{W} + \mathbf{A}\mathbf{B})$, which does not introduce extra inference costs. Note that it is inevitable that the absorption process will introduce additional quantization errors. To counteract this, following (He et al., 2017; Nagel et al., 2020; Hubara et al., 2020), we perform reconstruction sequentially rather than in parallel, which enables us to account for the quantization error stemming from the previous layers.

## 4.3 Efficiency Discussion

**Reassembly efficiency.** Our adaptive channel reassembly stands out for its efficiency, mainly attributed to its gradient-free nature, which excludes the need for backward propagation. The main source of computational expense of our method comes from the channel assembly, which requires the calculation of pairwise distances. Fortunately, the utilization of efficient bipartite soft matching eliminates the need to compute distances for every pair of channels, enhancing the efficiency. For the gradient-based error correction, the reduced number of parameters significantly lowers its optimization cost, rendering it more efficient than directly adjusting the quantized weights.

**Inference efficiency.** The inference overhead of channel disassembly and assembly is small for two reasons. 1) recent studies (Xiao et al., 2023; Wei et al., 2023) have revealed that activation outliers are often concentrated in specific channels across various inputs. This property is also reflected in similar channels for assembly as well. Therefore, we are able to pre-calculate the channel indices for disassembly and assembly using a small number of calibration data, significantly reducing runtime overhead. 2) Both channel disassembly and assembly can be implemented efficiently if the previous layer $l - 1$ is a linear layer. Please refer to Appendix B for more details. In cases where the preceding layer $l - 1$ is a non-linear layer, such as a layer normalization (Ba et al., 2016), we introduce additional disassembly and assembly layers that are designed to decompose and aggregate

Table 1: Performance comparisons of different methods for weights and activations quantization on LLaMA-1 model family. PPL denotes the perplexity.

| Model | #Bits | Method | PPL ↓ | | | Accuracy (%) ↑ | | | | | |
|---|---|---|---|---|---|---|---|---|---|---|---|
| | | | WikiText2 | C4 | Avg. | PIQA | ARC-e | ARC-c | HellaSwag | Winogrande | Avg. |
| LLaMA-1-7B | W16A16 | - | 5.68 | 7.08 | 6.38 | 77.37 | 52.48 | 41.38 | 72.99 | 66.93 | 62.23 |
| | W6A6 | SQ | 6.15 | 7.61 | 6.88 | 76.65 | 53.11 | 40.10 | 71.52 | 61.88 | 60.65 |
| | W6A6 | OS+ | 5.90 | - | - | 76.82 | 51.35 | 41.13 | 71.42 | 65.98 | 61.34 |
| | W6A6 | OmniQuant | 5.96 | 7.43 | 6.70 | 77.09 | 51.89 | 40.87 | 71.61 | 65.03 | 61.30 |
| | W6A6 | QLLM | 5.89 | 7.34 | **6.62** | 77.26 | 52.02 | 41.04 | 71.40 | 65.19 | **61.38** |
| | W4A8 | QLLM | 5.96 | 7.49 | 6.73 | 76.17 | 50.84 | 40.02 | 70.75 | 66.22 | 60.80 |
| | W4A4 | SQ | 52.85 | 104.35 | 78.60 | 49.80 | 30.40 | 25.80 | 27.40 | 48.00 | 36.28 |
| | W4A4 | LLM-QAT | - | - | - | 51.50 | 27.90 | 23.90 | 31.10 | 51.90 | 37.26 |
| | W4A4 | LLM-QAT+SQ | - | - | - | 55.90 | 35.50 | 26.40 | 47.80 | 50.60 | 43.24 |
| | W4A4 | OS+ | 40.32 | - | - | 62.73 | 39.98 | 30.29 | 44.39 | 52.96 | 46.07 |
| | W4A4 | OmniQuant | 11.26 | 14.51 | 12.89 | 66.15 | 45.20 | 31.14 | 56.44 | 53.43 | 50.47 |
| | W4A4 | QLLM | 9.65 | 12.29 | **10.97** | 68.77 | 45.20 | 31.14 | 57.43 | 56.67 | **51.84** |
| LLaMA-1-13B | W16A16 | - | 5.09 | 6.61 | 5.85 | 79.05 | 59.84 | 44.62 | 76.22 | 70.09 | 65.96 |
| | W6A6 | SQ | 5.50 | 7.03 | 6.27 | 77.80 | 56.36 | 42.58 | 75.11 | 68.11 | 63.99 |
| | W6A6 | OS+ | 5.37 | - | - | 78.29 | 56.90 | 43.09 | 75.09 | 69.22 | 64.52 |
| | W6A6 | OmniQuant | 5.28 | 6.84 | 6.06 | 78.40 | 57.28 | 42.91 | 75.82 | 68.27 | **64.54** |
| | W6A6 | QLLM | 5.28 | 6.82 | **6.05** | 77.91 | 57.70 | 42.92 | 75.02 | 69.14 | **64.54** |
| | W4A8 | QLLM | 5.33 | 6.91 | 6.12 | 78.29 | 57.03 | 42.75 | 74.46 | 68.35 | 64.18 |
| | W4A4 | SQ | 79.35 | 120.24 | 99.80 | 55.55 | 34.51 | 26.71 | 41.56 | 48.70 | 41.41 |
| | W4A4 | OS+ | 53.64 | - | - | 63.00 | 40.32 | 30.38 | 53.61 | 51.54 | 47.77 |
| | W4A4 | OmniQuant | 10.87 | 13.78 | 12.33 | 69.69 | 47.39 | 33.10 | 58.96 | 55.80 | 52.99 |
| | W4A4 | QLLM | 8.41 | 10.58 | **9.50** | 71.38 | 47.60 | 34.30 | 63.70 | 59.43 | **55.28** |
| LLaMA-1-30B | W16A16 | - | 4.10 | 5.98 | 5.04 | 80.09 | 58.92 | 45.39 | 79.21 | 72.77 | 67.28 |
| | W6A6 | SQ | 5.37 | - | - | 77.14 | 57.61 | 42.91 | 78.07 | 69.92 | 65.13 |
| | W6A6 | OS+ | 4.48 | - | - | 80.14 | 58.92 | 45.05 | 77.96 | 71.98 | 66.81 |
| | W6A6 | OmniQuant | 4.38 | 6.22 | 5.30 | 79.81 | 58.79 | 45.22 | 78.95 | 72.21 | **67.00** |
| | W6A6 | QLLM | 4.30 | 6.17 | **5.24** | 79.65 | 58.08 | 44.11 | 78.38 | 73.24 | 66.69 |
| | W4A8 | QLLM | 4.40 | 6.22 | 5.31 | 79.11 | 57.87 | 44.62 | 78.03 | 72.22 | 66.37 |
| | W4A4 | SQ | 399.65 | 245.87 | 322.76 | 50.16 | 28.11 | 26.71 | 31.97 | 51.14 | 37.62 |
| | W4A4 | OS+ | 112.33 | - | - | 67.63 | 46.17 | 34.30 | 54.32 | 52.64 | 51.01 |
| | W4A4 | OmniQuant | 10.33 | 12.49 | 11.41 | 71.21 | 49.45 | 34.47 | 64.65 | 59.19 | 55.79 |
| | W4A4 | QLLM | 8.37 | 11.51 | **9.94** | 73.83 | 50.67 | 38.40 | 67.91 | 58.56 | **57.87** |
| LLaMA-1-65B | W16A16 | - | 3.56 | 5.62 | 4.59 | 80.85 | 58.75 | 46.25 | 80.73 | 77.11 | 68.74 |
| | W6A6 | SQ | 4.00 | 6.08 | 5.04 | 77.97 | 54.67 | 44.62 | 77.51 | 72.61 | 65.48 |
| | W6A6 | OS+ | - | - | - | 79.67 | 55.68 | 45.22 | 78.03 | 73.95 | 66.51 |
| | W6A6 | OmniQuant | 3.75 | 5.82 | 4.79 | 81.01 | 58.12 | 46.33 | 79.91 | 75.69 | **68.21** |
| | W6A6 | QLLM | 3.73 | 5.80 | **4.77** | 80.14 | 57.79 | 45.05 | 79.74 | 74.59 | 67.46 |
| | W4A8 | QLLM | 3.78 | 8.82 | 6.30 | 80.14 | 58.59 | 46.42 | 79.71 | 74.66 | 67.90 |
| | W4A4 | SQ | 112.02 | 118.96 | 115.49 | 61.81 | 40.15 | 32.08 | 46.19 | 50.83 | 46.21 |
| | W4A4 | OS+ | 32.60 | - | - | 68.06 | 43.98 | 35.32 | 50.73 | 54.30 | 50.48 |
| | W4A4 | OmniQuant | 9.17 | 11.28 | 10.23 | 71.81 | 48.02 | 35.92 | 66.81 | 59.51 | 56.41 |
| | W4A4 | QLLM | 6.87 | 8.98 | **7.93** | 73.56 | 52.06 | 39.68 | 70.94 | 62.9 | **59.83** |

channels during runtime, with the channel indexes for decomposition and aggregation calculated offline using calibration data. The pseudo codes of channel disassembly and assembly during runtime can be found at Section D of supplementary material. Moreover, benefiting from our efficient kernel implemented by Triton (Tillet et al., 2019) and limited reassembly ratio searched by our adaptive strategy (See Figure C), the introduced inference cost is controlled within a small level.

## 5 EXPERIMENTS

**Models and datasets.** We apply QLLM to quantize the LLaMA-1 (Touvron et al., 2023a) and LLaMA-2 (Touvron et al., 2023b) families. To evaluate the performance of the quantized LLM, we report the zero-shot accuracy on various benchmarks, including PIQA (Bisk et al., 2020), ARC (Clark et al., 2018), HellaSwag (Zellers et al., 2019), and WinoGrande (Sakaguchi et al., 2021). Additionally, we evaluate the perplexity, a key indicator of a model's generative performance that correlates significantly with zero-shot outcomes, on WikiText2 (Merity et al., 2017), PTB (Marcus et al., 1993) and C4 (Raffel et al., 2020).

**Quantization settings.** In alignment with prior research (Dettmers et al., 2022; Shao et al., 2023), we use per-channel weight quantization and per-token activation quantization. Following (Shao et al., 2023; Liu et al., 2023), we quantize all weights and intermediate activations, with the exception of the Softmax output probability, which is maintained at full precision. Following OmniQuant (Shao et al., 2023), we focus on 4- and 6-bit weights and activations quantization. Additionally, we also explore 4-bit weights and 8-bit activations quantization, aiming for hardware-

friendly configurations while maintaining high performance. We exclude 8-bit quantization as SmoothQuant (Xiao et al., 2023) is able to achieve lossless performance.

**Compared methods.** We compare our QLLM with several state-of-the-art (SOTA) PTQ quantization methods, such as OmniQuant (Shao et al., 2023), SmoothQuant (SQ) (Xiao et al., 2023), Outlier Suppression+ (OS+) (Wei et al., 2023) and recent QAT method LLM-QAT (Liu et al., 2023). For fair comparisons, we reproduce SmoothQuant and Outlier Suppression+ with per-channel weight quantization and per-token activation quantization.

**Implementation details.** Following OmniQuant (Shao et al., 2023), we construct the calibration set with 128 randomly sampled sequences from WikiText2, each with a sequence length of 2048. QLLM begins by applying channel reassembly prior to all linear projection layers, excluding the attention output projection layer, followed by performing error correction on the resulting model. The rank $r$ of the introduced low-rank parameters is set to 4, and these parameters are trained for 10 epochs with a mini-batch size of 1. We carry out the reconstruction using 4 Attention-FFN blocks. AdamW (Loshchilov & Hutter, 2019) with a linear learning rate decay scheduler is used following (Yao et al., 2022). The learning rate is set to $5 \times 10^{-4}$ in most experiments; for LLaMA-2-70B, it is set to $1 \times 10^{-4}$. All training experiments are conducted on a single NVIDIA A100 80G GPU. We use the Language Model Evaluation Harness toolbox (Gao et al., 2021) for evaluation.

## 5.1 MAIN RESULTS

We report the results on LLaMA-1 and LLaMA-2 families in Table 1, and Table A in Appendix. Note that W6A6 has limited hardware support in real-world applications. However, our QLLM still demonstrates performance benefits in these settings, consistently surpassing OmniQuant in terms of lower perplexity across all models on both WikiText2 and C4 and achieving comparable accuracy on 5 zero-shot tasks. Remarkably, with W4A8 quantization, our method incurs only a minimal performance reduction. While the absolute performance gains with 6-bit quantization might seem modest, this is partly due to the less pronounced effect of activation outliers at this bitwidth. When focusing on extremely low-bitwidth quantization (*i.e.*, 4-bit), activation outliers serve as the performance bottleneck, thereby highlighting the importance of suppressing the outliers. In this case, our QLLM achieves significantly higher zero-shot accuracy and much lower perplexity than the contenders. For example, QLLM quantized 4-bit LLaMA-1-65B outperforms OmniQuant counterpart by an average of 3.42% in accuracy across five zero-shot tasks. Remarkably, for LLaMA-7B, our QLLM even surpasses the QAT method, LLM-QAT + SQ, by 8.6% on the average accuracy, which strongly demonstrates the efficacy of our QLLM.

Table 2: Perplexity results of different components in channel reassembly. "CD" stands for channel disassembly. "CA" represents channel assembly. "CP" indicates channel pruning. "Adaptive" refers to the adaptive strategy. "$\gamma$" is the channel expansion ratio.

| CD | CA | CP | Adaptive | $\gamma$ | LLaMA-1-13B | | | |
| --- | --- | --- | --- | --- | --- | --- | --- | --- |
| | | | | | WikiText2 | PTB | C4 | Avg. |
| ✓ | | | | 0.00 | 189.35 | 539.59 | 303.45 | 344.13 |
| ✓ | | | | 0.01 | 8.31 | 14.44 | 10.74 | 11.16 |
| ✓ | | | | 0.03 | 8.01 | 13.52 | 10.27 | 10.60 |
| ✓ | | | | 0.05 | 7.85 | 13.38 | 10.13 | 10.45 |
| ✓ | | | | 0.07 | 7.81 | 13.35 | 10.11 | **10.42** |
| ✓ | ✓ | | | 0.01 | 8.68 | 15.16 | 11.12 | 11.65 |
| ✓ | ✓ | | | 0.03 | 8.72 | 14.99 | 11.03 | **11.58** |
| ✓ | ✓ | | | 0.05 | 8.95 | 15.34 | 11.29 | 11.86 |
| ✓ | ✓ | | | 0.07 | 9.39 | 15.98 | 11.84 | 12.40 |
| ✓ | | ✓ | | 0.01 | 8.98 | 16.34 | 11.37 | **12.23** |
| ✓ | | ✓ | | 0.03 | 9.51 | 18.29 | 12.7 | 13.50 |
| ✓ | | ✓ | | 0.05 | 9.60 | 18.11 | 13.4 | 13.70 |
| ✓ | | ✓ | | 0.07 | 11.23 | 21.61 | 19.79 | 17.54 |
| ✓ | ✓ | - | ✓ | - | 8.41 | 14.38 | 10.58 | **11.12** |

Table 3: Inference throughput comparisons using a 2048-token segment on RTX 3090 GPUs: 1x GPU for LLaMA-1-7B and 2x GPUs for LLaMA-1-13B.

| Model | Method | Throughput (tokens/s) |
| --- | --- | --- |
| LLaMA-1-7B | FP16 | 3252 |
| | W8A8 | 5676 |
| | W4A16 | 5708 |
| | W4A4 | 6667 |
| | QLLM | 6385 |
| LLaMA-1-13B | FP16 | 1910 |
| | W8A8 | 3179 |
| | W4A16 | 2026 |
| | W4A4 | 3873 |
| | QLLM | 3730 |

## 5.2 ABLATION STUDIES

**Effect of different components in channel reassembly.** To show the effectiveness of the diverse components involved in channel reassembly, we apply different methods with our efficient error correction to yield 4-bit LLaMA-13B and show the results in Table 2. For channel disassembly, we determine $\theta$ by exploring different channel expansion ratios $\gamma$. We observe that our method with channel disassembly significantly surpasses the counterpart that does not utilize it. With the

increasing expansion ratio $\gamma$, the performance of the quantized model can be further improved. These results strongly show that channel disassembly is able to make activations more quantization-friendly by decomposing the outlier channels.

Furthermore, by incorporating channel assembly, our method manages to preserve the original channel count with little performance drop. In comparison to channel pruning, our channel assembly leads to lower information loss, thereby achieving much better performance, especially at higher $\gamma$. Rather than determining $\theta$ using a predefined expansion ratio, our method, equipped with an adaptive strategy, is capable of autonomously finding optimal $\theta$, resulting in near-lossless performance compared to the approach utilizing only channel disassembly. The resulting expansion ratios for different layers are shown in Figure C of the Appendix.

Table 4: Comparisons between efficient error correction (EEC) and tuning quantized weights directly (TQW) for 4-bit LLaMA-1-65B. "OOM" indicates out of memory.

| #Attn-FFN Block | Method | WikiText2 | PTB | C4 | Avg. | Training Time (GPU Hours) | GPU Memory (GB) |
|---|---|---|---|---|---|---|---|
| 1 | TQW | 6.34 | 17.61 | 9.56 | 11.17 | 12.16 | 30.84 |
| 1 | EEC | 8.31 | 13.77 | 10.76 | 10.95 | **7.79** | **19.00** |
| 2 | TQW | 6.25 | 11.18 | 8.56 | 8.66 | 12.13 | 52.45 |
| 2 | EEC | 7.62 | 11.47 | 9.39 | 9.49 | **7.79** | **28.60** |
| 4 | TQW | - | - | - | - | - | OOM |
| 4 | EEC | 6.87 | 11.36 | 8.98 | 9.07 | **7.77** | **47.71** |

**Effect of efficient gradient-based error correction.** After channel reassembly, we implement our QLLM to produce 4-bit LLaMA-7B models with our efficient gradient-based error correction (EEC) and tuning quantized weights directly (TQW) outlined in Section 4.2 to further improve the performance of quantized LLMs and show the results in Table 4. Compared with TQW which tunes all quantized weights, EEC focuses on learning a small set of low-rank weights, which significantly reduces training costs and GPU memory usage while delivering comparable performance. Moreover, the reduced GPU memory demand allows EEC to quantize LLaMA-1-65B on a single 24GB consumer-grade GPU, such as the NVIDIA RTX 4090, a task that is not feasible with TQW. Due to the page limited, we put more results in Section L of the supplementary material.

**Inference efficiency.** To assess the inference efficiency of our channel reassembly technique, we measure the inference speed of QLLM on NVIDIA RTX 3090 GPUs. We employ W4A4 kernels from QUIK (Ashkboos et al., 2023) codebase. We also conduct a comparative analysis using weight quantization only, utilizing CUDA kernels from AutoGPTQ[1]. As shown in Table 3, our 4-bit QLLM only incurs 4% additional cost relative to W4A4 but achieves a notable $1.96\times$ speedup over FP16. Notably, our channel reassembly strategy substantially mitigates losses attributed to quantizing outliers (see Table E), with only a slight extra computational overhead. For the detailed inference cost of channel disassembly and assembly, please refer to Section N of the supplementary material.

## 6 CONCLUSION AND FUTURE WORK

In this paper, we have proposed an accurate and efficient post-training quantization approach for low-bit LLMs, dubbed QLLM. The core of our QLLM lies in a novel adaptive channel reassembly paradigm that effectively addresses activation outliers, a pivotal factor contributing to the performance bottleneck in quantizing LLMs. The key idea involves reallocating outlier magnitudes to other channels, accomplished through a process of channel disassembly followed by assembly. We have further proposed a quantization-aware, parameter-efficient fine-tuning strategy that leverages calibration data to compensate for the information loss resulting from quantization. Extensive experiments on LLaMA model series have demonstrated the promising performance and training efficiency of QLLM. In terms of limitations, our proposed channel reassembly involves introducing additional operations to decompose and aggregate channels during runtime, thereby incurring additional inference costs. A potential solution to improve inference efficiency is to explore kernel fusing (Wang et al., 2010), aiming to fuse disassembly, assembly and layer normalization into a single operator. Another way is to aggregate more similar or unimportant channels (Sun et al., 2023) than those disassembled to achieve higher speedup.

---

[1]https://github.com/PanQiWei/AutoGPTQ

ACKNOWLEDGMENTS

We sincerely thank Shenghu Jiang for his help in implementing the efficient Triton kernel.

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

# Appendix

## A  MORE DETAILS ABOUT BIPARTITE SOFT MATCHING

As mentioned in Section 4.1.2, we use bipartite soft matching (Bolya et al., 2023) to determine which channels to aggregate efficiently. Support that we want to aggregate $T - 1$ channels. The step-by-step bipartite soft matching algorithm is shown as follows:

1. Divide the channels into two sets $\mathbb{A}$ and $\mathbb{B}$, each of approximately equal size.
2. For each channel in $\mathbb{A}$, construct an edge to its most similar counterpart in $\mathbb{B}$.
3. Select the $T - 1$ most similar edges.
4. Aggregate the channels that remain connected, according to Eq. (4).
5. Concatenate the two sets to form the assembled channel set.

## B  MORE DETAILS ABOUT THE EFFICIENT IMPLEMENTATION FOR CHANNEL REASSEMBLY

As mentioned in Section 4.3, the channel disassembly and assembly can be implemented efficiently if the previous layer $l - 1$ is a linear layer. Specifically, let $\mathbf{W}^{l-1} \in \mathbb{R}^{C \times M}$ be the weights of preceding linear layer, where $C$ and $M$ denotes the input and output channel number for layer $l-1$, respectively. For channel disassembly, we enlarge the output channels of the preceding linear layer weights by:

$$\mathbf{W}^{l-1}_{:i} = \begin{cases} \mathbf{W}^{l-1}_{:i} & \text{if } i \leq M - 1 \\ \frac{\mathbf{w}^{l-1}_{:M}}{T}, & \text{otherwise,} \end{cases} \tag{A}$$

and adjust the input channels of the current layer's weight by:

$$\mathbf{W}^{l}_{i:} = \begin{cases} \mathbf{W}^{l}_{i:} & \text{if } i \leq M - 1 \\ \mathbf{W}^{l}_{M:}, & \text{otherwise.} \end{cases} \tag{B}$$

Similarly, for channel assembly, suppose that we are aggregating channel $j$ to channel $i$. Then, channel assembly can be implemented by reducing the output weight channels of the preceding linear layer $l - 1$ by:

$$\mathbf{W}^{l-1}_{:i} = \frac{\mathbf{W}^{l-1}_{:i} + \mathbf{W}^{l-1}_{:j}}{2}, \tag{C}$$

and adjusting the input channels of the current layer's weight $l$ by:

$$\mathbf{W}^{l}_{i:} = \mathbf{W}^{l}_{i:} + \mathbf{W}^{l}_{j:}. \tag{D}$$

## C  ALGORITHM OF ADAPTIVE CHANNEL REASSEMBLY

We summarize our proposed adaptive channel reassembly in Algorithm 1.

## D  PSEUDO-CODES OF CHANNEL DISASSEMBLY AND ASSEMBLY

We show the PyTorch style pseudo-codes of channel disassembly and assembly during runtime in Figure A.

## E  MORE RESULTS ON LLAMA-2 FAMILY

We provide additional results for the LLaMA-2 family in Table A. The observations from these results are consistent with the phenomena identified in the LLaMA-1 family. Note that the varied performance of OmniQuant on W6A6 and W4A4 for LLaMA-2-70B can be attributed to the architecture of LLaMA-2-70B, which employs grouped-query attention (Ainslie et al., 2023) where each

---

**Algorithm 1:** Algorithm of Adaptive Channel Reassembly for one layer in LLM.

---

**Input:** Input activation $\mathbf{x} \in \mathbb{R}^M$, linear layer weight $\mathbf{W} \in \mathbb{R}^{M \times N}$, grid search iteration $P$.

Set $\mathcal{L}^*$ to $\infty$.

**Function** `ReAssembly(θ)`:

    // Channel disassembly

    Calculate the total sub-channels number by $n = \sum_{i=1}^{M} \lceil \max(|\mathbf{x}_i|)/\theta \rceil$.

    Perform channel disassembly using Eq. (3).

    // Channel assembly

    Find the $n$ most similar channel pairs using bipartite soft matching with the distance metric in Eq. (5).

    Perform channel assembly using Eq. (4).

    **return** ReAssembly Error $\mathcal{L}$ using Eq. (6);

Calculate the max value of each channel $\mathbf{m} \in \mathbb{R}^M$.

**for** $p \in \{1, 2, \ldots, P\}$ **do**

    Calculate the threshold by $\theta = \min(\mathbf{m}) + \frac{p}{P} \cdot (\max(\mathbf{m}) - \min(\mathbf{m}))$.

    $\mathcal{L}$=`ReAssembly(θ)`;

    **if** $\mathcal{L} < \mathcal{L}^*$ **then**

        $\mathcal{L}^* \leftarrow \mathcal{L}, \theta^* \leftarrow \theta$.

Adopt the final channel reassembly using the found $\theta^*$: `ReAssembly(θ*)`.

**return** One reassembled layer of LLM.

---

Table A: Performance comparisons of different methods for weights and activations quantization on LLaMA-2 model family.

| Model | #Bits | Method | PPL ↓ | | | Accuracy (%) ↑ | | | | | |
|---|---|---|---|---|---|---|---|---|---|---|---|
| | | | WikiText2 | C4 | Avg. | PIQA | ARC-e | ARC-c | HellaSwag | Winogrande | Avg. |
| LLaMA-2-7B | W16A16 | - | 5.47 | 6.97 | 6.22 | 76.82 | 53.62 | 40.53 | 72.87 | 67.25 | 62.22 |
| | W6A6 | SQ | 6.37 | 7.84 | 7.11 | 75.57 | 53.62 | 39.93 | 71.76 | 66.14 | 61.40 |
| | W6A6 | OS+ | - | - | - | 76.22 | 52.74 | 40.70 | 71.89 | 65.19 | 61.35 |
| | W6A6 | OmniQuant | 5.87 | 7.48 | 6.68 | 76.77 | 52.90 | 40.61 | 71.86 | 64.09 | 61.25 |
| | W6A6 | QLLM | 5.72 | 7.31 | **6.52** | 77.48 | 52.99 | 39.33 | 71.38 | 65.98 | **61.43** |
| | W4A8 | QLLM | 5.91 | 7.50 | 6.71 | 76.11 | 51.73 | 39.33 | 71.27 | 65.59 | 60.81 |
| | W4A4 | SQ | 101.77 | 93.21 | 97.49 | 60.17 | 35.23 | 27.13 | 37.08 | 49.57 | 41.84 |
| | W4A4 | OS+ | - | - | - | 63.11 | 39.10 | 28.84 | 47.31 | 51.3 | 45.93 |
| | W4A4 | OmniQuant | 14.61 | 18.39 | 16.50 | 65.94 | 43.94 | 30.80 | 53.53 | 55.09 | 49.86 |
| | W4A4 | QLLM | 11.75 | 13.26 | **12.51** | 67.68 | 44.40 | 30.89 | 58.45 | 56.59 | **51.60** |
| LLaMA-2-13B | W16A16 | - | 4.88 | 6.47 | 5.68 | 78.84 | 57.91 | 44.28 | 76.63 | 69.85 | 65.50 |
| | W6A6 | SQ | 5.19 | 6.77 | 5.98 | 78.29 | 57.41 | 43.86 | 75.02 | 66.93 | 64.30 |
| | W6A6 | OS+ | - | - | - | 78.29 | 59.13 | 43.34 | 75.37 | 67.56 | 64.74 |
| | W6A6 | OmniQuant | 5.14 | 6.74 | 5.94 | 78.56 | 57.11 | 43.60 | 75.36 | 68.35 | 64.60 |
| | W6A6 | QLLM | 5.08 | 6.71 | **5.90** | 78.78 | 58.29 | 43.77 | 75.10 | 68.43 | **64.87** |
| | W4A8 | QLLM | 5.17 | 6.78 | 5.98 | 78.67 | 57.11 | 41.89 | 75.33 | 68.75 | 64.35 |
| | W4A4 | SQ | 29.82 | 44.08 | 36.95 | 62.30 | 40.28 | 30.72 | 42.24 | 49.96 | 45.10 |
| | W4A4 | OS+ | - | - | - | 64.47 | 41.46 | 32.17 | 59.30 | 51.38 | 49.76 |
| | W4A4 | OmniQuant | 12.28 | 14.64 | 13.46 | 69.80 | 47.22 | 33.79 | 59.34 | 55.49 | 53.13 |
| | W4A4 | QLLM | 9.09 | 11.13 | **10.11** | 70.46 | 48.48 | 34.39 | 62.80 | 55.41 | **54.31** |
| LLaMA-2-70B | W16A16 | - | 3.32 | 5.52 | 4.42 | 81.01 | 59.68 | 47.95 | 80.87 | 76.95 | 69.29 |
| | W6A6 | SQ | 3.69 | 5.88 | 4.79 | 79.87 | 57.32 | 45.65 | 79.01 | 74.03 | 67.18 |
| | W6A6 | OS+ | - | - | - | 79.33 | 59.09 | 47.18 | 79.46 | 75.06 | 68.02 |
| | W6A6 | OmniQuant* | 3.71 | 5.91 | 4.81 | 80.20 | 60.27 | 46.84 | 80.55 | 76.01 | **68.77** |
| | W6A6 | QLLM | 3.55 | 5.76 | **4.66** | 80.63 | 59.01 | 45.99 | 79.64 | 75.37 | 68.13 |
| | W4A8 | QLLM | 3.6 | 5.76 | 4.68 | 80.79 | 58.59 | 47.44 | 79.42 | 75.77 | 68.40 |
| | W4A4 | SQ | 26.01 | 34.61 | 30.31 | 64.09 | 41.84 | 32.00 | 54.21 | 51.07 | 48.64 |
| | W4A4 | OS+ | - | - | - | 66.16 | 42.72 | 34.90 | 56.93 | 52.96 | 50.73 |
| | W4A4 | OmniQuant* | 41.10 | 54.33 | 47.72 | 52.99 | 31.14 | 23.89 | 33.88 | 52.01 | 38.78 |
| | W4A4 | QLLM | 7.00 | 8.89 | **7.95** | 74.27 | 50.59 | 37.2 | 71.62 | 59.43 | **58.62** |

* indicates no learnable equivalent transformation (Shao et al., 2023) on queries, keys, values, or attention output due to incompatibility with grouped-query attention (Ainslie et al., 2023) in LLaMA-2-70B model.

group of queries shares a single key and value head. Such architecture makes the learnable equivalent transformation in OmniQuant incompatible with grouped-query attention. In W6A6 settings, the impact of activation outliers is relatively minor, enabling partial learnable equivalent transforma-

```python
def channel_disassembly(x, num_split):
    """
    x: input with shape of [batch, tokens, channels]
    num_split: the number of sub-channels for each channel with shape of [channels]
    """

    B, N, C = x.shape
    x = x.view(B * N, C)
    scaling = 1.0 / num_split # compute the scaling factor of each channel
    x = x / scaling # scale each channel
    x = torch.repeat_interleave(x, num_split, dim=1) # perform channel decomposition
    C = x.shape[1]
    x = x.view(B, N, C)
    return x

def channel_assembly(x, src_idx, dst_idx):
    """
    x: input with shape of [batch, tokens, channels]
    src_idx: the channel index that will be merged in set A with shape of [#num_merged_channels]
    dst_idx: the channel index that will be merged in set B with shape of [#num_merged_channels]
    """
    B, N, C = x.shape
    ori_src_idx = torch.arange(0, C, 2, device=x.device) # get index for set A
    ori_dst_idx = torch.arange(1, C, 2, device=x.device) # get index for set B
    src, dst = x[..., ori_src_idx], x[..., ori_dst_idx] # divide the channels into two sets A and B
    src_C = src.shape[-1] # get the channel number in set A
    dst_C = dst.shape[-1] # get the channel number in set B

    # A mask that indicates whether a channel is merged
    channel_mask = torch.ones(C, device=x.device, dtype=x.dtype)
    m_idx = ori_src_idx[src_idx]
    channel_mask[m_idx] = 0.0

    n, t1, c = src.shape
    sub_src = src.gather(dim=-1, index=src_idx.expand(n, t1, r)) # get channels that will be merged
            in set A
    dst = dst.scatter_reduce(-1, dst_idx.expand(n, t1, r), sub_src, reduce=mode) # merge channels
    src = src.view(B, N, src_C, 1)
    dst = dst.view(B, N, dst_C, 1)

    # concat set A and set B
    if src_C == dst_C:
        merged_x = torch.cat([src, dst], dim=-1).view(B, N, C)
    else:
        merged_x = torch.cat([src[..., :-1, :], dst], dim=-1).view(
            B, N, src_C + dst_C - 1
        )
        merged_x = torch.cat([merged_x, src[..., -1, :].reshape(B, N, 1)], dim=-1).view(
            B, N, src_C + dst_C
        )
    # remove the merged channels
    merged_x = merged_x.index_select(-1, (channel_mask != 0).nonzero().squeeze())
    return merged_x
```

Figure A: PyTorch style pseudo codes of channel disassembly and assembly during runtime.

tions to suffice in maintaining performance. However, in the W4A4 settings, the effect of activation outliers becomes more prominent. Under these conditions, the partial learnable equivalent transformation is insufficient to address the outlier issue, leading to notably poorer performance. Notably, our QLLM significantly outperforms the state-of-the-art post-training quantization (PTQ) methods, demonstrating a substantial margin of improvement in 4-bit quantization. For example, QLLM quantized 4-bit LLaMA-2-70B outperforms SmoothQuant counterpart by an average of 7.89% on the accuracy, which shows the promising results of our method.

# F    MORE RESULTS ON CHAT MODELS

To demonstrate the generalization ability of our QLLM on chat models, we apply QLLM to quantize LLaMA-2-7B-Chat and LLaMA-2-13B-Chat to 4-bit. These models are instruction-tuned and optimized for dialogue use cases. We include the concurrent state-of-the-art quantizaiton method, OmniQuant, for comparisons. We use GPT-4 to assess the performance of the quantized models on

a set of 80 sample questions in Vicuna benchmark (Chiang et al., 2023). To eliminate the potential position bias (Zheng et al., 2023), we conducted the comparisons in both orders (a *vs.*b and b *vs.*a) for each pair, amounting to a total of 160 trials. From Table B, our QLLM consistently achieves much better performance than OmniQuant.

Table B: Performance comparisons between QLLM and OmniQuant for chat models.

| Model | Case | Former Win | Tie | Former Lost |
|---|---|---|---|---|
| LLaMA-2-7B-Chat | QLLM vs. OmniQuant | **137** | 19 | 4 |
| LLaMA-2-13B-Chat | QLLM vs. OmniQuant | **116** | 24 | 20 |

## G   MORE RESULTS IN TERMS OF CHANNEL REASSEMBLY

To further show the effectiveness of our channel reassembly (CR), we compare the average block-wise reconstruction error across the entire network before and after applying CR and show the results on a calibration set with 128 randomly selected 2048-token segments from WikiText2 in Table C. The results clearly demonstrate that using CR significantly lowers the reconstruction error, and thus improves the performance of the quantized models.

Table C: Block-wise reconstruction error before and after channel reassembly (CR).

| Model | Method | Reconstruction Error |
|---|---|---|
| LLaMA-1-7B | w/o CR | 4.71 |
| LLaMA-1-7B | w/ CR | **2.74** |
| LLaMA-1-13B | w/o CR | 7.67 |
| LLaMA-1-13B | w/ CR | **1.71** |

## H   MORE RESULTS IN TERMS OF CHANNEL DISASSEMBLY ONLY

To further demonstrate the effectiveness of channel disassembly (CD), we apply CD without efficient error correction (EEC) to obtain 4-bit LLaMA-1-13B and show the results in Table D. We observe that the absence of both CD and EEC leads to a significant decline in the performance of the quantized model. Notably, using CD alone substantially reduces the performance degradation associated with quantization. Moreover, increasing the channel expansion ratio $\gamma$ further improves the model's performance, which strongly shows the benefits of using CD to decompose the outlier channels. By incorporating both CD and EEC, the performance improvement is even more pronounced, underscoring the efficacy of EEC in conjunction with CD.

Table D: Perplexity results of channel disassembly (CD) with and without efficient error correction (EEC). "$\gamma$" is the channel expansion ratio. We report the perplexity of W4A4 LLaMA-1-13B on WikiText2 (Merity et al., 2017), PTB (Marcus et al., 1993) and C4 (Raffel et al., 2020).

| CD | EEC | $\gamma$ | WikiText2 | PTB | C4 | Avg. |
|---|---|---|---|---|---|---|
| | | - | 1702.34 | 1853.58 | 1159.41 | 1571.78 |
| ✓ | | 0.01 | 19.34 | 45.36 | 23.25 | 29.32 |
| ✓ | ✓ | 0.01 | **8.31** | **14.44** | **10.74** | **11.16** |
| ✓ | | 0.03 | 12.11 | 24.73 | 14.38 | 17.07 |
| ✓ | ✓ | 0.03 | **8.01** | **13.52** | **10.27** | **10.60** |
| ✓ | | 0.05 | 11.4 | 23.53 | 13.62 | 16.18 |
| ✓ | ✓ | 0.05 | **7.85** | **13.38** | **10.13** | **10.45** |
| ✓ | | 0.07 | 11.13 | 23.47 | 13.45 | 16.02 |
| ✓ | ✓ | 0.07 | **7.81** | **13.35** | **10.11** | **10.42** |

## I    MORE COMPARISONS WITH OTHER OUTLIER HANDLING METHODS

To further show the effectiveness of channel reassembly, we compare our method with previous outlier handling methods which employ gradient-free methods to learn mathematically equivalent transformations. For fair comparisons, we do not apply efficient error correction. From Table E, all methods exhibit comparable performance at 6-bit quantization. However, for 4-bit quantization, channel reassembly significantly surpasses other methods by a large margin, particularly for larger models.

Table E: Performance comparisons of our channel reassembly (CR) with previous outlier handling methods across five zero-shot tasks.

| Model | #Bits | Method | PIQA | ARC-e | ARC-c | HellaSwag | Winogrande | Avg. |
|---|---|---|---|---|---|---|---|---|
| | W6A6 | SQ | 76.65 | 53.11 | 40.10 | 71.52 | 61.88 | 60.65 |
| | W6A6 | OS+ | 76.82 | 51.35 | 41.13 | 71.42 | 65.98 | **61.34** |
| LLaMA-1-7B | W6A6 | CR | 76.88 | 52.31 | 40.87 | 71.37 | 64.33 | 61.15 |
| | W4A4 | SQ | 49.80 | 30.40 | 25.80 | 27.40 | 48.00 | 36.28 |
| | W4A4 | OS+ | 62.73 | 39.98 | 30.29 | 44.39 | 52.96 | 46.07 |
| | W4A4 | CR | 66.92 | 42.55 | 32.34 | 54.31 | 50.04 | **49.23** |
| | W6A6 | SQ | 77.80 | 56.36 | 42.58 | 75.11 | 68.11 | 63.99 |
| | W6A6 | OS+ | 78.29 | 56.90 | 43.09 | 75.09 | 69.22 | **64.52** |
| LLaMA-1-13B | W6A6 | CR | 78.02 | 56.69 | 42.41 | 74.70 | 70.01 | 64.37 |
| | W4A4 | SQ | 55.55 | 34.51 | 26.71 | 41.56 | 48.70 | 41.41 |
| | W4A4 | OS+ | 63.00 | 40.32 | 30.38 | 53.61 | 51.54 | 47.77 |
| | W4A4 | CR | 67.57 | 43.77 | 31.48 | 60.78 | 56.04 | **51.93** |

## J    MORE RESULTS IN TERMS OF THE EFFICIENT ERROR CORRECTION ONLY

Using EEC only without our channel reassembly results in suboptimal performance as it suffers from activation outlier issues. To demonstrate this, we applied EEC only to quantize LLaMA-1-7B to 4-bit, using the same training settings as our QLLM but with varying numbers of calibration samples. From Table F, even with an increased amount of calibration data, the performance of the EEC only significantly lags behind our QLLM. These results strongly demonstrate the effectiveness of channel reassembly in addressing activation outliers, thereby substantially improving performance.

Table F: Performance comparisons with different methods under various numbers of calibration samples. We report the perplexity of W4A4 LLaMA-1-7B on WikiText2 (Merity et al., 2017), PTB (Marcus et al., 1993) and C4 (Raffel et al., 2020).

| Method | #Samples | WikiText2 | PTB | C4 | Avg. |
|---|---|---|---|---|---|
| EEC | 128 | 16.64 | 38.58 | 28.33 | 27.85 |
| EEC | 256 | 14.94 | 37.70 | 33.62 | 28.75 |
| EEC | 512 | 12.35 | 29.39 | 31.59 | 24.44 |
| QLLM | 128 | **9.65** | **16.56** | **12.29** | **12.83** |

## K    MORE RESULTS IN TERMS OF TUNING QUANTIZED WEIGHTS ONLY

The effectiveness of TQW is highly dependent on our channel reassembly. To demonstrate this, we applied TQW only to quantize LLaMA-1-7B to 4-bit using the same training settings as QLLM and show the results in Table G. The results clearly indicate that the absence of our adaptive channel reassembly results in significantly reduced performance for TQW. This underscores the vital role of channel reassembly in addressing activation outliers and thus improving model performance.

## L    MORE COMPARISONS BETWEEN EFFICIENT ERROR CORRECTION AND TUNING QUANTIZED WEIGHTS DIRECTLY

To further show the effectiveness of our efficient error correction (EEC), we conduct more comparisons between EEC and tuning quantized weights (TQW) directly on small model and report the re-

Table G: Performance comparisons with different methods. We report the perplexity of 4-bit LLaMA-1-7B on WikiText2 (Merity et al., 2017), PTB (Marcus et al., 1993) and C4 (Raffel et al., 2020). "CR" denotes our adaptive channel reassembly.

| Method | WikiText2 | PTB | C4 | Avg. |
|---|---|---|---|---|
| TQW w/o CR | 13.13 | 42.81 | 32.07 | 29.34 |
| TQW w/ CR | **8.90** | **14.75** | **11.63** | **11.76** |

sults in Table H. The results show that employing EEC not only maintains comparable performance but also markedly improves training speed and significantly reduces GPU memory usage over TQW. It is worth noting that there is a trade-off between GPU memory (*i.e.*, #Attn-FFN blocks) and performance. Leveraging EEC even allows us to perform reconstruction for 16 Attention-FFN blocks simultaneously, thereby significantly improving performance while preserving a similar training speed and a reasonable increase in GPU memory.

Table H: Perplexity comparisons between efficient error correction (EEC) and tuning quantized weights directly (TQW) for 4-bit LLaMA-1-7B. "OOM" indicates out of memory.

| #Attn-FFN Block | Method | WikiText2 | PTB | C4 | Avg. | Training Time (GPU Hours) | GPU Memory (GB) |
|---|---|---|---|---|---|---|---|
| - | CR | 14.12 | 25.30 | 16.58 | 18.67 | - | - |
| 1 | TQW | 10.10 | 16.19 | 12.95 | 13.08 | 1.49 | 10.9 |
| | EEC | 11.21 | 19.29 | 14.06 | 14.85 | **1.06** | **7.95** |
| 2 | TQW | 9.74 | 14.79 | 11.68 | 12.07 | 1.49 | 17.62 |
| | EEC | 10.61 | 18.56 | 13.64 | 14.27 | **1.05** | **11.62** |
| 4 | TQW | 8.90 | 14.75 | 11.63 | 11.76 | 1.48 | 30.95 |
| | EEC | 9.65 | 16.56 | 12.29 | 12.83 | **1.05** | **18.91** |
| 8 | TQW | - | - | - | - | - | OOM |
| | EEC | 9.18 | 14.98 | 11.63 | 11.93 | **1.05** | **33.50** |
| 16 | TQW | - | - | - | - | - | OOM |
| | EEC | 9.13 | 14.95 | 11.60 | 11.89 | **1.05** | **62.70** |

## M EFFECT OF THE WEIGHT MERGING IN EFFICIENT ERROR CORRECTION

As explained in Section 4.2, for $\text{quant}(\mathbf{X})\text{quant}(\mathbf{W}) + \text{quant}(\mathbf{X})\mathbf{AB}$, the low-rank weights $\mathbf{A}$ and $\mathbf{B}$ bring not only additional inference overhead due to the matrix multiplication between the full-precision $\mathbf{AB}$ and $\text{quant}(\mathbf{X})$ but also extra storage burden. To address this, we perform weight merging by $\text{quant}(\mathbf{W} + \mathbf{AB})$ after the reconstruction, which effectively avoids overhead but introduces additional quantization error. For 4-bit quantization, we empirically observe that merging the low-rank weights into the frozen weights using $\text{quant}(\mathbf{W} + \mathbf{AB})$ does not lead to an increase in outliers. This finding is supported by the notably low MSE levels for channel-wise $P_{99}$, $P_{999}$, and maximum/minimum values before and after the weight merging process across in Table I. Moreover, our weight merging only leads to small quantization error, as shown in Table J. Note that even small deviations can aggregate throughout the network, leading to the performance drop. To address this, as shown in Section 4.2, we further employ sequential reconstruction to mitigate errors from previous layers, resulting in only a negligible performance drop. To demonstrate this, we compare the performance of QLLM with and without the weight merging. From Table K, the weight merging only leads to a slight increase in perplexity.

Table I: The maximum MSE of channel-wise $P_{99}$, $P_{999}$, and maximum/minimum values before and after the merging process across all layers of 4-bit LLaMA-1-7B.

| $\text{MSE}_{P_{99}}$ | $\text{MSE}_{P_{999}}$ | $\text{MSE}_{\max}$ | $\text{MSE}_{\min}$ |
|---|---|---|---|
| $5.42 \times 10^{-6}$ | $3.48 \times 10^{-6}$ | $7.11 \times 10^{-6}$ | $8.71 \times 10^{-6}$ |

Table J: The maximum MSE of channel-wise $P_{99}$, $P_{999}$, and maximum/minimum values before and after the merging process across all layers of 4-bit LLaMA-1-7B.

| Model | MSE of Weight Merging |
|---|---|
| LLaMA-1-7B | $4.04 \times 10^{-7}$ |
| LLaMA-1-13B | $3.64 \times 10^{-7}$ |

Table K: Effect of the weight merging (WM) in the efficient error correction. We report the perplexity on WikiText2 (Merity et al., 2017), PTB (Marcus et al., 1993) and C4 (Raffel et al., 2020).

| Model | Method | WikiText2 | PTB | C4 | Avg. |
|-------|--------|-----------|-----|-----|------|
| LLaMA-1-7B | w/o WM | 9.35 | 15.93 | 11.93 | 12.40 |
| LLaMA-1-7B | w/ WM | 9.65 | 16.56 | 12.29 | 12.83 |
| LLaMA-1-13B | w/o WM | 8.29 | 13.69 | 10.40 | 10.79 |
| LLaMA-1-13B | w/ WM | 8.41 | 14.38 | 10.58 | 11.12 |

## N  MORE RESULTS REGARDING INFERENCE EFFICIENCY

Following (Guo et al., 2020; Wang et al., 2020), we use Bit-Operation (BOP) count to measure the theoretical inference complexity of our QLLM. From Table L, our 8-bit QLLM incurs only a marginal increase in BOPs when compared to the INT8 model but substantially lower than those of the FP16 counterpart, which shows the efficiency of our method.

Table L: Bit-Operation (BOP) count comparisons of different models. We report the results of LLaMA-1-7B with a mini-batch size of 1. "$L$" denotes the sequence length.

| $L$ | 256 | 512 | 1024 | 2048 |
|-----|-----|-----|------|------|
| FP16 | 875.52T | 1,766.40T | 3,604.48T | 7,493.12T |
| INT8 | 231.58T | 467.64T | 952.56T | 1976.16T |
| QLLM | 231.58T+1.07M | 467.64T+2.14M | 952.56T+4.28M | 1976.16T+8.56M |

We further show the inference time of channel disassembly and assembly of our QLLM in Table M. From the results, channel disassembly results in additional inference costs due to the extra channels. These additional channels often don't align with GPU-friendly multiples like 32 or 64, leading to less efficient GPU use. Using our channel assembly maintains the original channel count, ensuring better GPU utilization and mitigating the extra inference costs from disassembly. As a result, the quantized models with both channel disassembly and assembly achieve higher throughput compared to the ones with disassembly only, which demonstrates the necessity of channel assembly.

Table M: Inference throughput (tokens/s) comparisons of different models. The throughput is measured with a 2048-token segment on NVIDIA RTX 3090 GPUs: 1x GPU for LLaMA-1-7B and 2x GPUs for LLaMA-1-13B. "CD" stands for channel disassembly. "CA" represents channel assembly. "Adaptive" refers to the adaptive strategy. "$\gamma$" is the channel expansion ratio. "OOM" indicates out of memory.

| Model | Method | CD | CA | Adaptive | $\gamma$ | Inference Throughput (tokens/s) |
|-------|--------|-----|-----|----------|----------|-------------------------------|
| | FP16 | | | | - | 3252 |
| | W8A8 | | | | - | 5676 |
| | W4A16 | | | | - | 5708 |
| | W4A4 | | | | - | 6667 |
| | W4A4 | ✓ | | | 0.01 | 6322 |
| LLaMA-1-7B | W4A4 | ✓ | | | 0.05 | 6315 |
| | W4A4 | ✓ | | | 0.1 | 6310 |
| | W4A4 | ✓ | ✓ | | 0.01 | 6365 |
| | W4A4 | ✓ | ✓ | | 0.05 | 6334 |
| | W4A4 | ✓ | ✓ | | 0.1 | 6318 |
| | W4A4 | ✓ | ✓ | ✓ | - | 6385 |
| | FP16 | | | | - | 1910 |
| | W8A8 | | | | - | 3179 |
| | W4A16 | | | | - | 2026 |
| | W4A4 | | | | - | 3873 |
| | W4A4 | ✓ | | | 0.01 | 3728 |
| LLaMA-1-13B | W4A4 | ✓ | | | 0.05 | 3725 |
| | W4A4 | ✓ | | | 0.1 | 3678 |
| | W4A4 | ✓ | ✓ | | 0.01 | 3731 |
| | W4A4 | ✓ | ✓ | | 0.05 | 3728 |
| | W4A4 | ✓ | ✓ | | 0.1 | 3681 |
| | W4A4 | ✓ | ✓ | ✓ | - | 3730 |

## O    MORE RESULTS REGARDING TRAINING EFFICIENCY

We assess the training efficiency of our method in comparison to OmniQuant on a single NVIDIA A100 80G GPU. The GPU training hours for both methods are presented in Table N. The results reveal that the training cost of our QLLM can be up to $1.93\times$ faster than OmniQuant, showing the exceptional training efficiency of our QLLM.

Table N: The training time (GPU Hours) comparisons of our QLLM with OmniQuant.

| Method | OmniQuant | QLLM |
|---|---|---|
| LLaMA-2-7B | 1.98 | **1.05** |
| LLaMA-2-13B | 3.46 | **1.79** |
| LLaMA-2-70B | 14.52 | **9.05** |

## P    EFFECT OF DIFFERENT CALIBRATION SETS

We apply QLLM to yield 4-bit LLaMA-7B using different calibration sets and report the results in Table O. From the results, we observe that the choices of calibration set have a minor effect, as the performance remains relatively consistent across different sets. For fair comparisons, following OmniQuant (Shao et al., 2023), we use WikiText2 as a calibration set by default.

Table O: Effect of different calibration sets. We report the perplexity ↓ of W4A4 LLaMA-7B on WikiText2 (Merity et al., 2017), PTB (Marcus et al., 1993) and C4 (Raffel et al., 2020).

| | | Evaluation Set | | | |
|---|---|---|---|---|---|
| | | WikiText2 | PTB | C4 | Average |
| | WikiText2 | 9.65 | 16.56 | 12.29 | 12.83 |
| Calibration Set | PTB | 10.73 | 13.89 | 12.44 | **12.35** |
| | C4 | 10.56 | 17.44 | 12.08 | 13.36 |

## Q    EFFECT OF DIFFERENT NUMBERS OF CALIBRATION SAMPLES

We apply QLLM to yield 4-bit LLaMA-7B using different numbers of calibration samples from WikiText2 and show the results in Table P. The results reveal a positive correlation between the performance of the quantized model and the number of calibration samples, indicating that utilizing more samples generally leads to better performance. This trend underscores the importance of the calibration phase, where leveraging a larger sample pool can provide a more comprehensive representation of the data distribution, enabling more accurate quantization. However, it is also imperative to consider the computational and memory overhead associated with an increasing number of calibration samples. There is an inherent trade-off between achieving higher model performance and maintaining computational efficiency.

Table P: Effect of different # calibration samples. We report the perplexity ↓ of W4A4 LLaMA-7B on WikiText2 (Merity et al., 2017), PTB (Marcus et al., 1993) and C4 (Raffel et al., 2020).

| #Samples | WikiText2 | PTB | C4 | Average |
|---|---|---|---|---|
| 16 | 10.72 | 18.43 | 13.66 | 14.27 |
| 32 | 10.12 | 17.35 | 12.84 | 13.44 |
| 64 | 10.15 | 16.23 | 12.26 | 12.88 |
| 128 | 9.65 | 16.56 | 12.29 | 12.83 |
| 256 | 9.60 | 15.75 | 11.79 | **12.38** |

## R    MORE RESULTS ABOUT THE EXPANSION RATIOS OF THE QUANTIZED LLM

In this section, we illustrate the detailed expansion ratios for the input activations of different layers in the 4-bit LLaMA-1-7B and LLaMA-1-13B obtained by our adaptive strategy in Figures B and C. From the results, our adaptive strategy allocates higher expansion ratios to the shallower MSA

layers and to the deeper down projection layer in the FFN, which indicates that these layers possess a greater number of outliers. To substantiate this observation, we further plot the channel-wise maximum and minimum values for the input activations across different layers in Figure D. These visual representations further underscore the effectiveness of our adaptive strategy in identifying and addressing the presence of outliers in different layers.

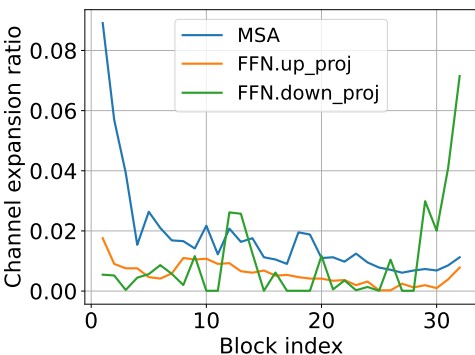

Figure B: An illustration of the searched expansion ratios using our adaptive strategy for 4-bit LLaMA-1-7B.

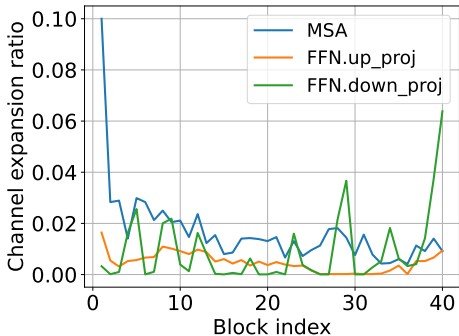

Figure C: An illustration of the searched expansion ratios using our adaptive strategy for 4-bit LLaMA-1-13B.

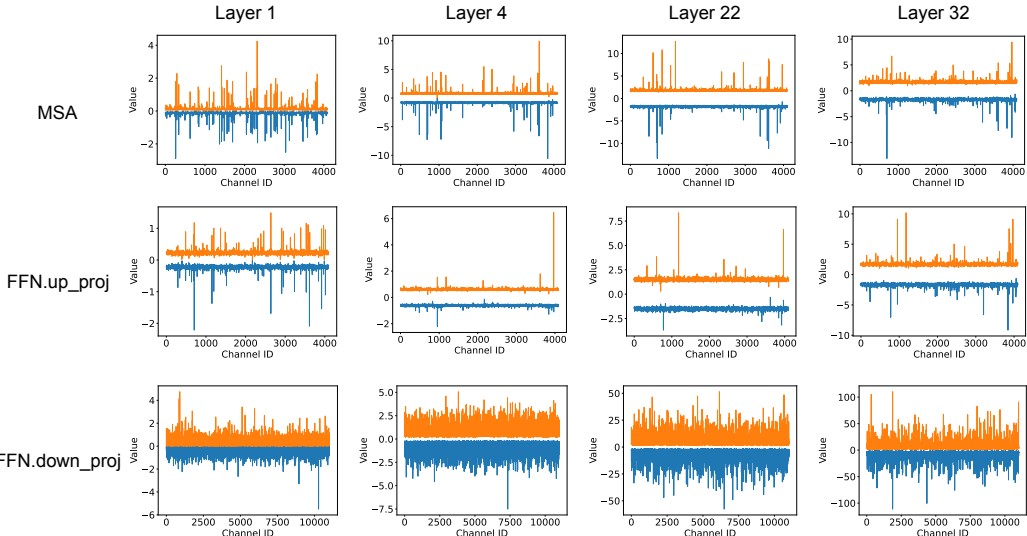

Figure D: An illustration of the channel-wise maximum and minimum input activation values for the MSA, up projection and down projection layers in FFN of different blocks in LLaMA-1-13B.

