# OpenReview forum: "QLLM: Accurate and Efficient Low-Bitwidth Quantization for Large Language Models"
_ICLR.cc/2024/Conference — ICLR 2024 poster_

### Official Review · Reviewer_rXfg · 2023-10-26

**Soundness:** 3 good
**Presentation:** 3 good
**Contribution:** 3 good
**Rating:** 6
**Confidence:** 4

**Summary:**

In this paper, the authors proposed QLLM for low-bit weight-and-activation quantization of LLMs. QLLM consists of three techniques: channel disassembly to reduce outlier magnitude and channel assembly to maintain channel numbers, an adaptive strategy to find the optimal expansion ratios, and a low-rank fine-tuning method to reduce quantization error. Experiments show that the proposed method outperforms existing quantization methods, especially under low-bit setting (W4A4).

**Strengths:**

1. The paper is generally well-written and easy to follow.
2. The proposed method outperforms existing ones under low-bit weight and activation quantization (W4A4).
3. The ablation study is comprehensive, showing the effectiveness of each components (e.g., channel assembly, adaptive expansion rate, etc.)
4. The efficient fine-tuning method seems like an efficient way to restore performance.

**Weaknesses:**

1. The proposed method has a significant advantage *only* under very low-bit settings like W4A4. However, under W4A4, all methods (including QLLM) cannot achieve a reasonable accuracy. Take Table 1 as an example, the W4A4 LLaMA-65B accuracy of QLLM is only 59.83% (average), which is even lower than the FP16 accuracy of LLaMA-7B, which is 62.23%. The huge drop in accuracy makes the setting impractical. On the other hand, the proposed method does not outperform existing work like OmniQuant under higher precisions (e.g., W6A6). Therefore, it is questionable whether the proposed method has a practical advantage.
2. The proposed disassembly and assembly method will lead to inevitable inference overhead when there is LayerNorm or activation functions, since extra layers are inserted in the the forward graph (also confirmed in Table 4, 27% overhead compared to INT8 under context length 256, which is negligible). Furthermore, it is unclear what kind of configurations (e.g., disassembly ratios) are used for the INT8 setting in Table 4, since there is no accuracy number reported under this setting. If we use a larger disassembly ratio, will the overhead be larger?
3. It seems from Table 6 does directly updating the quantized weights (i.e., QAT) is always better than the LoRA tuning. Does it mean if we have enough GPUs for parallelized fine-tuning (e.g., FSDP), we should use QAT for the best accuracy?

**Questions:**

Please see the weakness section.

---

> ### Author Response · Authors · 2023-11-19
> **Response to Reviewer rXfg**
>
> Thanks for your constructive comments.
>
> **Q1.** The proposed method has a significant advantage only under very low-bit settings like W4A4. However, under W4A4, all methods (including QLLM) cannot achieve a reasonable accuracy. On the other hand, the proposed method does not outperform existing work like OmniQuant under higher precisions (e.g., W6A6). Therefore, it is questionable whether the proposed method has a practical advantage.
>
> **A1**. We would like to emphasize that the advantages of our QLLM extend beyond low-bit settings. It is crucial to comprehensively evaluate LLM performance using a variety of metrics, including perplexity (PPL) and zero-shot accuracy, across different benchmarks. Notably, in W6A6 quantization settings, QLLM consistently outperforms OmniQuant in terms of lower PPL on both WikiText2 and C4 across all models, as shown in Tables 1 and A.
>
> Additionally, our method falls within the post-training quantization settings, where we need to quantize a pre-trained LLM using a small amount of calibration data with little computational overhead. This setup is in line with mainstream LLM quantization methods (Xiao et al., 2023, Wei et al., 2023, Shao et al., 2023) and inherently faces some performance constraints due to the smaller scale of data and training. Notably, in W4A4 settings, QLLM significantly outperforms the concurrent state-of-the-art quantization method, OmniQuant, by a large margin, as shown in Tables 1 and A. Moreover, our QLLM is complementary to large-scale fine-tuning to further improve performance. For example, by using 40 Attention-FFN Blocks for reconstruction and more calibration samples, our 4-bit quantized LLaMA-1-65B achieves **61.04% average accuracy on five zero-shot tasks**, which is **1.21%** higher than the result in the previous submission. Given the time constraints of this rebuttal period, we believe there is potential for more performance gains.
>
> **Q2**. The proposed disassembly and assembly method will lead to inevitable inference overhead when there are LayerNorm or activation functions, since extra layers are inserted in the forward graph (also confirmed in Table 4, 27% overhead compared to INT8 under context length 256, which is negligible).
>
> **A2**. We would like to correct a misconception about the inference cost associated with our QLLM. The actual additional inference cost, when compared to INT8 at a context length of 256, is only 15% rather than 27%, as shown in Table 4 in the initial submission. To further reduce this overhead, we implement our channel disassembly and assembly operation using Triton [A], optimizing these processes for enhanced efficiency. Additionally, the channel expansion ratios determined by our adaptive strategy are kept within a small range (as detailed in Q3), ensuring that the increase in inference overhead remains small. For detailed inference overhead, please refer to Q1 of the general response.
>
> **Reference**
>
> [A] Triton: an intermediate language and compiler for tiled neural network computations. MAPL 2019.
>
> **Q3**. It is unclear what kind of configurations (e.g., disassembly ratios) are used for the INT8 setting in Table 4. If we use a larger disassembly ratio, will the overhead be larger?
>
> **A3**. In Table 4, we set the channel expansion ratio $\gamma$ to 0.01, which was chosen because it reflects the average channel expansion ratio across various layers. While a larger $\gamma$ value does lead to increased overhead, our efficient implementation ensures that this increase remains small, as demonstrated in Table I of the general response. Importantly, as illustrated in Figure A, the channel expansion ratios determined by our adaptive strategy do not exceed 0.1. This cap acts as an effective upper bound, ensuring that the additional inference overhead is both predictable and constrained within manageable limits.
>
> **Q4**. It seems from Table 6 does tuning quantized weights directly (TQW) is always better than the efficient error correction (EEC). Does it mean if we have enough GPUs for parallelized fine-tuning (e.g., FSDP), we should use TQW for the best accuracy?
>
> **A4**. While TQW might be preferable in situations where ample GPU resources are available, our EEC shines in resource-limited settings, delivering performance that is on par with TQW. A notable example of EEC's efficiency is its ability to quantize the LLaMA-1-65B model using only a single 24GB consumer-grade GPU, such as the NVIDIA RTX 4090.
> This efficiency makes EEC especially beneficial for researchers facing resource constraints, markedly improving the accessibility of quantization technology for LLMs. For a comparison that underscores the benefits of EEC over TQW, please refer to Q3 of the general response.

---

> > ### Comment · Reviewer_rXfg · 2023-11-19
> >
> > Thanks to the authors for the informative rebuttal, which addresses part of my concerns.
> >
> > Despite the performance advantage gap being kind of small under W6A6, I would like to raise my ratings to 6 given other merits.

---

> > > ### Author Response · Authors · 2023-11-19
> > > **Thanks for your feedback**
> > >
> > > Dear Reviewer rXfg,
> > >
> > > Thank you for your feedback! We are pleased to address your questions and greatly appreciate your reviews, which play a crucial role in improving our work.
> > >
> > > Best regards,
> > >
> > > Authors of Paper #54

---

### Official Review · Reviewer_wUVT · 2023-10-28

**Soundness:** 3 good
**Presentation:** 2 fair
**Contribution:** 2 fair
**Rating:** 6
**Confidence:** 4

**Summary:**

In this paper, the authors propose QLLM, a precise and efficient post-training quantization method specifically designed for Large Language Models (LLMs). They address the challenge of activation outliers by introducing a gradient-free channel reassembly technique that redistributes the magnitudes of outlier channels across all channels. This ensures a more balanced activation range, facilitating accurate quantization and improving the performance of quantized LLMs. The authors also introduce channel assembly to maintain the original channel count by merging similar channels. Additionally, they propose an adaptive strategy to determine the optimal number of disassembled channels for each layer, based on minimizing the reassembly error. The proposed QLLM method achieves accurate quantized models efficiently, as demonstrated by extensive experiments on LLaMA-1 and LLaMA-2 datasets. For example, QLLM outperforms the previous state-of-the-art method by 7.89% on average accuracy across five zero-shot tasks for the 4-bit LLaMA-2-70B model, trained within 10 hours on a single A100-80G GPU.

**Strengths:**

- The concepts of CHANNEL DISASSEMBLY and CHANNEL ASSEMBLY proposed in this paper appear to be novel. The findings presented in Table 2 provide evidence for the effectiveness of CD and CA.
- The methodology described in this paper is straightforward and comprehensible.
- The authors have conducted thorough experiments across various settings, which is commendable.

**Weaknesses:**

## Major Concern:
- While I acknowledge the efficiency of CHANNEL DISASSEMBLY and CHANNEL ASSEMBLY in the form of $y=xW^{l-1}W^{l}$, as explained in "MORE DETAILS ABOUT THE EFFICIENT IMPLEMENTATION FOR CHANNEL REASSEMBLY", I have reservations regarding its applicability to scenarios involving multiple inputs $X=\\{x_1,x_2,...,x_K\\}\in\mathbb{R}^{K\times M}$ and a non-linear transformation $y=\phi(xW^{l-1})W^{l}$, where $\phi(\cdot)$ represents a normalization layer followed by an activation function like GELU. Despite the authors' claim that "the channel indexes for decomposition and aggregation are calculated offline using calibration data, which only introduces a minor extra overhead", I remain unconvinced that the overhead is negligible.

## Minor Comments:
- It would be beneficial to provide a proof demonstrating that the approximation error introduced by Equation (4), in conjunction with the subsequent quantization error, is indeed smaller than the quantization error resulting from direct quantization.
- I am also curious about the potential occurrence of outliers when employing $quant(W+AB)$ in the equation $y=quant(x)quant(W+AB)$. Additionally, I would appreciate further insights into the quantization error between $quant(X)quant(W+AB)$ and $quant(X)(quant(W)+AB)$.

**Questions:**

- I am interested in whether the proposed CHANNEL DISASSEMBLY and CHANNEL ASSEMBLY methods can be extended to handle multiple inputs, specifically in the case of $Y=XW$ where $X\in\mathbb{R}^{batchsize\times M}$. I have some concerns regarding the effectiveness of Equations (3) and (4) on $X\in\mathbb{R}^{batchsize\times M}$ if the outliers differ among $X_{i,:}$, as the derivation relies on the approximation of the summation of scalars. Additionally, the authors mention that "with the channel indexes for decomposition and aggregation calculated offline using calibration data". Did the authors imply that the outliers often occur at the same index across different inputs? Otherwise, the effectiveness of the proposed methods may vary.

- The notation of $\beta$ in Equation (3) appears to correspond to $\lfloor\frac{\min(X)}{\alpha}\rceil$.

- It would be beneficial to include the baseline results of $\gamma=0$ in Table 2.

- According to Table 6 and the abstract, the training time of efficient error correction is reported to be around 1 hour, while the total time cost is 10 hours, suggesting that the adaptive CD/CA takes approximately 9 hours. Furthermore, the baseline result of tuning quantized weights directly (TQW) with 4 Attn-FFN Blocks seems to be quite strong, as it only takes one and a half hours with much smaller GPU memory overhead compared to the 16 Attn-FFN Blocks EEC results, and it even achieves better performance. Could the authors provide further clarification on this matter?

- Did the FP results reported in Table 4 refer to FP32 or FP16 results?

- I am curious about the baseline results of tuning quantized weights directly (TQW) with EFFICIENT GRADIENT-BASED ERROR CORRECTION. Did this setting outperform the proposed CD/CA + EEC?

- Could the authors provide pseudo-codes to provide more detailed explanations of the "additional disassembly and assembly layers that are designed to decompose and aggregate channels during runtime"?

- Could the authors further elaborate on why OmniQuant consistently performs well on W6A6 but performs extremely poorly on LLaMA-2-70B with W4A4?

- It is worth noting that "DIFFERENT NUMBERS OF CALIBRATION SAMPLES" play a crucial role in performance. Did the authors follow the same setting as other baseline methods?

---

> ### Author Response · Authors · 2023-11-19
> **Response to Reviewer wUVT (Part 1)**
>
> Thanks for your constructive comments.
>
> **Q1**. I have reservations regarding its applicability to scenarios involving multiple inputs $\mathbf{X}= \\{ \mathbf{x}_1,\mathbf{x}_2,...,\mathbf{x}_K \\} \in\mathbb{R}^{K\times M}$ and a non-linear transformation $\mathbf{y}=\phi(\mathbf{x}\mathbf{W}^{l-1})\mathbf{W}^{l}$, where $\phi(\cdot)$ represents a normalization layer followed by an activation function like GELU. Despite the authors' claim that "the channel indexes for decomposition and aggregation are calculated offline using calibration data, which only introduces a minor extra overhead", I remain unconvinced that the overhead is negligible.
>
> **A1**. Please refer to Q1 of the general response for the inference overhead of channel reassembly. With our efficient Triton kernel implementation, our channel reassembly only incurs **4% additional runtime overhead** compared with W4A4.
>
> **Q2**. It would be beneficial to provide a proof demonstrating that the approximation error introduced by Equation (4), in conjunction with the subsequent quantization error, is indeed smaller than the quantization error resulting from direct quantization.
>
> **A2**. To further show the effectiveness of our channel reassembly (CR), we compare the average block-wise reconstruction error across the entire network before and after applying CR and show the results on a calibration set with 128 randomly selected 2048-token segments from WikiText2 in Table VIII. The results clearly demonstrate that using CR significantly lowers the reconstruction error, and thus improves the performance of the quantized models. We have included these results and the corresponding discussions in Section G of the supplementary material.
>
> Table VIII. Block-wise reconstruction error before and after channel reassembly (CR).
> | Model       | Method | Reconstruction Error |
> | ----------- | :----: | :------------------: |
> | LLaMA-1-7B  | w/o CR |         4.71         |
> | LLaMA-1-7B  | w/ CR  |       **2.74**       |
> | LLaMA-1-13B | w/o CR |         7.67         |
> | LLaMA-1-13B | w/ CR  |       **1.71**       |
>
> **Q3**. I am also curious about the potential occurrence of outliers when employing $\mathrm{quant}(\mathbf{W}+\mathbf{A}\mathbf{B})$ in the equation $\mathbf{Y}=\mathrm{quant}(\mathbf{X})\mathrm{quant}(\mathbf{W}+\mathbf{A}\mathbf{B})$.
>
> **A3**. We empirically observe that merging the low-rank weights into the frozen weights using $\mathrm{quant}(\mathbf{W}+\mathbf{A}\mathbf{B})$ does not lead to an increase in outliers. This finding is supported by the high cosine similarity (**nearly 0.999**) of the channel-wise maximum/minimum values before and after the merging process.
>
> **Q4**. I would appreciate further insights into the quantization error between $\mathrm{quant}(\mathbf{X})\mathrm{quant}(\mathbf{W}+\mathbf{A}\mathbf{B})$ and $\mathrm{quant}(\mathbf{X})(\mathrm{quant}(\mathbf{W})+\mathbf{A}\mathbf{B})$.
>
> **A4**. As explained in Section 4.2, for $\mathrm{quant}(\mathbf{X})(\mathrm{quant}(\mathbf{W})+\mathbf{A}\mathbf{B})$, the low-rank weights $\mathbf{A}$ and $\mathbf{B}$ bring not only additional inference overhead due to the matrix multiplication between the full-precision $\mathbf{A}\mathbf{B}$ and $\mathrm{quant}(\mathbf{X})$ but also extra storage burden. To address this, we perform weight merging by $\mathrm{quant}(\mathbf{W}+\mathbf{A}\mathbf{B})$ after the reconstruction, which effectively avoids overhead but introduces additional quantization error. For 4-bit quantization, our empirical findings show that the average cosine similarity between $\mathrm{quant}(\mathbf{W})+\mathbf{A}\mathbf{B}$ and $\mathrm{quant}(\mathbf{W}+\mathbf{A}\mathbf{B})$ across the network is **over 0.999**, indicating a small quantization error. Additionally, we further employ sequential reconstruction to mitigate errors from previous layers, resulting in only a negligible performance drop. To demonstrate this, we compare the performance of QLLM with and without the weight merging. From Table IX, the weight merging only leads to a slight increase in perplexity. We have included the above discussions and corresponding results in Section L of the supplementary material.
>
> Table IX. Effect of the weight merging (WM) in the efficient error correction. We report the perplexity on WikiText2, PTB and C4.
> | Model       | Method | WikiText2 |  PTB  |  C4   | Avg.  |
> | ----------- | :----: | :-------: | :---: | :---: | :---: |
> | LLaMA-1-7B  | w/o WM |   9.35    | 15.93 | 11.93 | 12.40 |
> | LLaMA-1-7B  | w/ WM  |   9.65    | 16.56 | 12.29 | 12.83 |
> | LLaMA-1-13B | w/o WM |   8.29    | 13.69 | 10.40 | 10.79 |
> | LLaMA-1-13B | w/ WM  |   8.41    | 14.38 | 10.58 | 11.12 |

---

> > ### Comment · Reviewer_wUVT · 2023-11-21
> > **Re: Response to Reviewer wUVT (Part 1)**
> >
> > I would like to seek clarification on a few points:
> >
> > 1. When referring to the "average block-wise reconstruction error," does this term indicate the mean squared error (MSE) computed before and after network quantization?
> >
> > 2. In my opinion, it would be more informative to include the differences in the P99, P999, and maximum values of the channel-wise maximum/minimum values before and after the merging process. This additional information could provide a more comprehensive understanding of the results.
> >
> > 3. I have noticed a potential inconsistency between the reported performance drop in Table VIII and the statement that "the average cosine similarity between $quant(W)+AB$ and $quant(W+AB)$ across the network is over 0.999." It appears that cosine similarity may not be an appropriate metric in this context. Could the authors please address this discrepancy?

---

> > > ### Author Response · Authors · 2023-11-22
> > > **Response to additional questions (Part 4)**
> > >
> > > **Q17**. When referring to the "average block-wise reconstruction error," does this term indicate the mean squared error (MSE) computed before and after network quantization?
> > >
> > > **A17**. Yes, the ``average block-wise reconstruction error’’ denotes the mean squared error (MSE) calculated between outputs of each Attention-FFN block before and after network quantization, averaged across the network.
> > >
> > > **Q18**. In my opinion, it would be more informative to include the differences in the P99, P999, and maximum values of the channel-wise maximum/minimum values before and after the merging process. This additional information could provide a more comprehensive understanding of the results.
> > >
> > > **A18**. Following your suggestion, we show the maximum MSE for channel-wise P99, P999, and maximum/minimum values before and after the weight merging process across all layers in Table XI. The notably low MSE levels support that our weight merging process does not contribute to an increase in outliers. We include the results in Section L in the supplementary material.
> > >
> > > Table XI. The maximum MSE of channel-wise $P99$, $P999$, and maximum/minimum values before and after the merging process across all layers of 4-bit LLaMA-1-7B.
> > > | MSE$_{P99}$ | MSE$_{P999}$ | MSE$_{\mathrm{max}}$ | MSE$_{\mathrm{min}}$ |
> > > | :--------------: | :---------------: | :---------------: | :------------: |
> > > | 5.42 $\times 10^{-6}$ | 3.48 $\times 10^{-6}$ | 7.11$\times 10^{-6}$ | 8.71 $\times 10^{-6}$ |
> > >
> > > **Q19**. I have noticed a potential inconsistency between the reported performance drop in Table VIII and the statement that "the average cosine similarity between $\mathrm{quant}(\mathbf{W})+\mathbf{A}\mathbf{B}$ and $\mathrm{quant}(\mathbf{W}+\mathbf{A}\mathbf{B})$ across the network is over 0.999." It appears that cosine similarity may not be an appropriate metric in this context. Could the authors please address this discrepancy?
> > >
> > > **A19**. The discrepancy can be attributed to the cumulative effect of even minor errors during forward propagation. While each individual layer shows a high degree of cosine similarity and thus a low average MSE as demonstrated in Table XII, these small deviations can aggregate throughout the network, leading to the observed performance drop in Table IX. We include the results in Section L in the supplementary material.
> > >
> > > Table XII. The average MSE between $\mathrm{quant}(\mathbf{W})+\mathbf{A}\mathbf{B}$ and $\mathrm{quant}(\mathbf{W}+\mathbf{A}\mathbf{B})$ across the network.
> > > | Model       | MSE of Weight Merging |
> > > | ------------- | :--------: |
> > > | LLaMA-1-7B  | 4.04 $\times 10^{-7}$ |
> > > | LLaMA-1-13B | 3.64 $\times 10^{-7}$ |

---

> ### Author Response · Authors · 2023-11-19
> **Response to Reviewer wUVT (Part 2)**
>
> **Q5**. I am interested in whether the proposed channel disassembly and channel assembly methods can be extended to handle multiple inputs. I have some concerns regarding the effectiveness of Equations (3) and (4) on $\mathbf{X}\in\mathbb{R}^{batchsize\times M}$ if the outliers differ among $\mathbf{X}_{i,:}$, as the derivation relies on the approximation of the summation of scalars. Did the authors imply that the outliers often occur at the same index across different inputs?
>
> **A5**. Our channel disassembly and assembly are indeed designed to work with different inputs, effectively identifying outliers and similar channels. As we have highlighted in Section 1, drawing on recent findings from studies (Dettmers et al., 2022; Xiao et al., 2023; Wei et al., 2023), activation outliers are often found to cluster within specific channels across a variety of inputs. This property is reflected in similar channels for channel assembly as well, which tend to exhibit consistent patterns irrespective of the input. Such uniformity allows for the identification of these channels with a small set of calibration data. We have included the discussions in Section 4.3 of the revised manuscript.
>
> **Q6**. The notation of $\beta$ in Eq. (2) appears to correspond to $\lfloor\frac{\min(X)}{\alpha}\rceil$.
>
> **A6**. We confirm that the notation of the zero-point $\beta$ in Eq. (2) is correct, which serves the purpose of offsetting the range of values to enable storage of quantized values in an unsigned integer format.
>
> **Q7**. It would be beneficial to include the baseline results of $\gamma=0$ in Table 2.
>
> **A7**. We would like to clarify that the baseline results for $\gamma=0$ have been included in the first row of Table 2. To enhance clarity and prevent any potential misunderstandings, we have updated the presentation of these results in our revised manuscript as follows.
>
> |  CD   |  CA   |  CP   | Adaptive | $\gamma$ | WikiText2 |  PTB   |   C4   |  Avg.  |
> | :---: | :---: | :---: | :------: | :------: | :-------: | :----: | :----: | :----: |
> |   ✓   |       |       |          |    0     |  189.35   | 539.59 | 303.45 | 344.13 |
>
> **Q8**. According to Table 6 and the abstract, the training time of efficient error correction is reported to be around 1 hour, while the total time cost is 10 hours, suggesting that the adaptive channel disassembly/channel assembly takes approximately 9 hours.
>
> **A8**. We would like to clarify a misunderstanding regarding the training time. The time reported in Table 6 is for the 4-bit LLaMA-1-7B model, while the abstract shows the training time for the 4-bit LLaMA-2-70B model. Additionally, as discussed in Section 4.3, our channel disassembly and assembly process is highly training efficient, largely due to its gradient-free design. To illustrate, the channel disassembly and assembly for the 4-bit LLaMA-1-7B require only **9.6 minutes**.
>
> **Q9**. The baseline result of tuning quantized weights directly (TQW) with 4 Attn-FFN Blocks seems to be quite strong, as it only takes one and a half hours with much smaller GPU memory overhead compared to the 16 Attn-FFN Blocks efficient error correction (EEC) results, and it even achieves better performance. Could the authors provide further clarification on this matter?
>
> **A9**. We would like to clarify that both TQW and EEC are training strategies to further improve the performance of the quantized LLMs **after adaptive channel reassembly**. While TQW shows strength in certain scenarios, EEC offers significant advantages, particularly in resource efficiency. EEC specifically targets learning a limited set of low-rank weights, thereby reducing training costs and GPU memory requirements. This approach enables EEC to quantize larger models like the LLaMA-1-65B on a single 24GB GPU, a task not feasible with TQW. For a detailed comparison of EEC's advantages over TQW, please see Q3 of the general response.
>
> **Q10**. Did the FP results reported in Table 4 refer to FP32 or FP16 results?
>
> **A10**. The FP results reported in Table 4 refer to FP16 results. We have revised our manuscript.
>
> **Q11**. I am curious about the baseline results of tuning quantized weights with efficient error correction (EEC). Did this setting outperform the proposed CD/CA + EEC?
>
> **A11**. Please refer to Q2 of the general response for the detailed results. Using EEC only without channel reassembly results in poor performance as it suffers from activation outlier issues.

---

> > ### Comment · Reviewer_wUVT · 2023-11-21
> > **Re: Response to Reviewer wUVT (Part 2)**
> >
> > Thank you for providing detailed responses. However, I still have a few remaining questions:
> >
> > 1. Regarding Equation (2), if the authors maintain the correctness of $\beta=-\lfloor\frac{\min(X)}{\alpha}\rceil$, it seems that the formulation of $X_q=quant(X)=\text{clamp}(\lfloor\frac{\min(X)}{\alpha}\rceil+\beta,0,2^{b}-1)$ would quantize most values to 0, assuming that the weights $X$ generally follow a zero mean Gaussian distribution. Could the authors please address this concern?
> >
> > 2. Additionally, I would appreciate it if the authors could provide the results of TQW without adaptive channel reassembly.

---

> ### Author Response · Authors · 2023-11-19
> **Response to Reviewer wUVT (Part 3)**
>
> **Q12**. Could the authors provide pseudo-codes to provide more detailed explanations of the "additional disassembly and assembly layers that are designed to decompose and aggregate channels during runtime"?
>
> **A12**. To provide a clearer understanding, we include the pseudo-codes of our disassembly and assembly during runtime in PyTorch style as follows:
>
> ```python
> def channel_disassembly(x, num_split):
> """
>     x: input with shape of [batch, tokens, channels]
>     num_split: the number of sub-channels for each channel with shape of [channels]
> """
>
>     B, N, C = x.shape
>     x = x.view(B * N, C)
>     scaling = 1.0 / num_split   # compute the scaling factor of each channel
>     x = x / scaling                    # scale each channel
>     x = torch.repeat_interleave(x, num_split, dim=1) # perform channel decomposition
>     C = x.shape[1]
>     x = x.view(B, N, C)
>     return x
> ```
>
> ``` python
> def channel_assembly(x, src_idx, dst_idx):
> """
>     x: input with shape of [batch, tokens, channels]
>     src_idx: the channel index that will be merged in set A with shape of [#num_merged_channels]
>     dst_idx: the channel index that will be merged in set B with shape of [#num_merged_channels]
> """
>     B, N, C = x.shape
>     ori_src_idx = torch.arange(0, C, 2, device=x.device)  # get index for set A
>     ori_dst_idx = torch.arange(1, C, 2, device=x.device)  # get index for set B
>     src, dst = x[..., ori_src_idx], x[..., ori_dst_idx]    # divide the channels into two sets A and B
>     src_C = src.shape[-1]    # get the channel number in set A
>     dst_C = dst.shape[-1]    # get the channel number in set B
>
>     # A mask that indicates whether a channel is merged
>     channel_mask = torch.ones(C, device=x.device, dtype=x.dtype)
>     m_idx = ori_src_idx[src_idx]
>     channel_mask[m_idx] = 0.0
>
>     n, t1, c = src.shape
>     sub_src = src.gather(dim=-1, index=src_idx.expand(n, t1, r))   # get channels that will be merged in set A
>     dst = dst.scatter_reduce(-1, dst_idx.expand(n, t1, r), sub_src, reduce=mode)  # merge channels
>     src = src.view(B, N, src_C, 1)
>     dst = dst.view(B, N, dst_C, 1)
>
>     # concat set A and set B
>     if src_C == dst_C:
>         merged_x = torch.cat([src, dst], dim=-1).view(B, N, C)
>     else:
>         merged_x = torch.cat([src[..., :-1, :], dst], dim=-1).view(
>             B, N, src_C + dst_C - 1
>         )
>         merged_x = torch.cat([merged_x, src[..., -1, :].reshape(B, N, 1)], dim=-1).view(
>             B, N, src_C + dst_C
>         )
>     # remove the merged channels
>     merged_x = merged_x.index_select(-1, (channel_mask != 0).nonzero().squeeze())
>     return merged_x
> ```
> We have included the above pseudo-codes in Section D of the supplementary material.
>
> **Q13**. Could the authors further elaborate on why OmniQuant consistently performs well on W6A6 but performs extremely poorly on LLaMA-2-70B with W4A4?
>
> **A13**. The varied performance of OmniQuant can be attributed to the architecture of LLaMA-2-70B, which employs grouped-query attention (Ainslie et al., 2023) where each group of queries shares a single key and value head. Such architecture makes the learnable equivalent transformation in OmniQuant incompatible with grouped-query attention. In W6A6 settings, the impact of activation outliers is relatively minor, enabling partial learnable equivalent transformations to suffice in maintaining performance. However, in the W4A4 settings, the effect of activation outliers becomes more prominent. Under these conditions, the partial learnable equivalent transformation is insufficient to address the outlier issue, leading to notably poorer performance. We have included the analysis in Section E of the appendix.
>
> **Q14**. Did the authors follow the same number of calibration samples as other baseline methods?
>
> **A14**. Yes, we use the same number of calibration samples as OmniQuant, as already mentioned in the implementation details of Section 5.

---

> > ### Author Response · Authors · 2023-11-21
> > **Follow-up on Rebuttal**
> >
> > Dear Reviewer wUVT
> >
> > Thank you once more for your efforts in reviewing our paper. We have made significant efforts to address your major concerns regarding the additional overhead of channel assembly. Should you have any remaining questions or concerns, please feel free to reach out to us.
> >
> > Best regards,
> >
> > Authors of #54

---

> ### Author Response · Authors · 2023-11-22
> **Response to additional questions (Part 3)**
>
> Thanks for your feedback.
>
> **Q15**. Regarding Equation (2), if the authors maintain the correctness of $\beta=-\lfloor\frac{\min(\mathbf{X})}{\alpha}\rceil$, it seems that the formulation of $\mathbf{X}_q=\mathrm{quant}(\mathbf{X})=\text{clamp}(\lfloor\frac{\min(\mathbf{X})}{\alpha}\rceil+\beta,0,2^{b}-1)$ would quantize most values to 0, assuming that the weights $\mathbf{X}$ generally follow a zero mean Gaussian distribution. Could the authors please address this concern?
>
> **A15**. We would like to clarify that the correct quantization function is $\mathbf{X}_q = \mathrm{quant}(\mathbf{X}) = \mathrm{clamp} \left( \lfloor \frac{\mathbf{X}}{\alpha} \rceil + \beta, 0, 2^b - 1 \right)$, as mentioned in Eq. (2). As a result, only the values of $\mathbf{X}$ extremely close to $\min(\mathbf{X})$ are quantized to zero. All other values of $\mathbf{X}$ are mapped within the range of $\left(0, 2^{b}-1 \right]$.
>
> **Q16**. Additionally, I would appreciate it if the authors could provide the results of TQW without adaptive channel reassembly.
>
> **A16**. We show the results of TQW only in Table X in Q5 of the general response. The results clearly indicate that the absence of our adaptive channel reassembly results in significantly reduced performance for TQW. This underscores the vital role of channel reassembly in addressing activation outliers and thus improving model performance. We include the results in Section J in the supplementary material.

---

### Official Review · Reviewer_KacJ · 2023-10-30

**Soundness:** 3 good
**Presentation:** 3 good
**Contribution:** 2 fair
**Rating:** 6
**Confidence:** 4

**Summary:**

This work, QLLM, mainly addresses activation outliers for quantizing LLMs. QLLM proposes to first break the outlier channels into several channels, and then merge similar channels to keep the original channel number (by bipartite soft matching). To decide how many channels an outlier channel needs to be broken into, QLLM conducts a grid search of the outlier threshold hyper-parameter $\theta$ using layer-wise reconstruction error as the objective. Then the channel-reassembled network is more suitable for activation quantization. Finally, QLLM adopts LoRA training with block-wise reconstruction error as the objective to restore performance. Experiments are conducted with LLaMA v1 and v2 models. Evaluations are conducted on WikiText, PTB, and C4 for PPL, and 5 zero-shot benchmarks for zero-shot accuracy. The algorithm performance experiments use the W6A6 and W4A4 settings, and the inference efficiency experiments use the W8A8 setting.

**Strengths:**

* This paper is well-written, well-organized, and easy to follow.
* Motivation: The activation outlier issue is important for quantizing activations in LLMs
* Reasonable method: Channel splitting is a reasonable method to address the activation outlier issue.

**Weaknesses:**

My major concern in this paper is whether its current evaluation can fully demonstrate its practicability.

**About the inference efficiency**
* While the algorithm perf. experiments are conducted with W6A6 and W4A4, inference efficiency experiments are only conducted with W8A8. I think implementing W4A4 kernels on some NVIDIA GPUs and demonstrating better efficiency-algo. perf. trade-off compared with W8A8 and existing W4A4 will make this paper stronger.
* Table 4 shows QLLM has inference overhead compared to methods without channel reassembly. Does this overhead come from the additional disassembly and assembly layers for non-linear layers? A thorough breakdown of this overhead for differently-sized models (larger models, more compressed models such as 4bit) will help illustrate whether the method is really practical.
* As the channel-assembly process introduces additional approximations, only applying the channel-disassembly process results in the best perplexity (which is shown in Table 2). So, to further demonstrate the necessity of applying channel assembly to keep the channel number fixed, the authors can show the inference efficiency when using the expanded channel number (only apply CD).

**About the algorithm performance**
* I'm curious how much influence the reassembly process still has as the training data amount goes up. I.e., do the authors compare with using only reconstruction-based LoRA tuning without CD, CA, and CP, under exactly the same training settings but maybe with more calibration data (e.g., 256 instead of 128).
* A suggestion instead of a weakness: To raise more interest in using this method, I recommend doing some evaluations on chat models.

**Questions:**

* Important points are listed in the weakness section.
* Sec 4.1.1 uses $\max(x_M)$ to determine the number of disassembly channels $T$ according to $\theta$, why don't we use $\max(|x_M|)$?
* I wonder whether the CD results in Table 2 use the LoRA tuning? I would like to see the results of using CD without LoRA tuning as a reference.

---

> ### Author Response · Authors · 2023-11-19
> **Response to Reviewer KacJ (Part 1)**
>
> Thanks for your constructive comments.
>
> **Q1**. I think implementing W4A4 kernels on some NVIDIA GPUs and demonstrating better efficiency-algo. perf. trade-off compared with W8A8 and existing W4A4 will make this paper stronger.
>
> **A1**. Thanks for your valuable suggestion. Please refer to Q1 of the general response for the inference time results.
>
> **Q2**. Does the overhead come from the additional disassembly and assembly layers for non-linear layers? A thorough breakdown of this overhead for differently-sized models (larger models, more compressed models such as 4bit) will help illustrate whether the method is really practical.
>
> **A2**. Yes, the additional inference overhead comes from the additional disassembly and assembly layers after the non-linear layers. Please refer to Q1 of the general response for a thorough breakdown of the inference overhead for differently-sized models. Additionally, we further show the inference overhead of QLLM for 4-bit LLaMA-2-70B. From Table V, our QLLM introduces only a **2% additional overhead** compared with 4-bit fixed-point baseline.
>
> Table V. Inference throughput (tokens/s) comparisons of different models. The throughput is measured with a 2048-token segment on 8 NVIDIA RTX 4090 GPUs.
> | Model | Method | Inference Throughput (tokens/s) |
> | - | - | :-: |
> | LLaMA-2-70B | FP16 | 824 |
> | LLaMA-2-70B | W8A8 | 1406 |
> | LLaMA-2-70B | W4A4 | 1677 |
> | LLaMA-2-70B | W4A4 | 1638 |
>
> **Q3**. To further demonstrate the necessity of applying channel assembly to keep the channel number fixed, the authors can show the inference efficiency when using the expanded channel number (only apply CD).
>
> **A3**. Please refer to Q1 of the general response for the inference efficiency with and without channel assembly. Our results show that applying both channel disassembly and assembly leads to higher throughput compared to using disassembly alone as it is more hardware-friendly.
>
> **Q4**. I'm curious how much influence the reassembly process still has as the training data amount goes up. I.e., do the authors compare with using only efficient error correction (EEC) without CD, CA, and CP, under exactly the same training settings but maybe with more calibration data (e.g., 256 instead of 128).
>
> **A4**. Please refer to Q2 of the general response for the detailed results. Using EEC only without channel reassembly results in poor performance as it suffers from activation outlier issues.
>
> **Q5**. A suggestion instead of a weakness: To raise more interest in using this method, I recommend doing some evaluations on chat models.
>
> **A5**. Following your suggestion, we apply QLLM to quantize LLaMA-2-7B-Chat and LLaMA-2-13B-Chat to 4-bit. We compared our models with the concurrent state-of-the-art quantization method, OmniQuant. We use GPT-4 to assess the performance of the quantized models on a set of 80 sample questions in the Vicuna benchmark [A]. To eliminate the potential position bias [B], we conducted the comparisons in both orders (a vs. b and b vs. a) for each pair, amounting to a total of 160 trials. From Table VI, our QLLM consistently achieves much better performance than OmniQuant, demonstrating the effectiveness of our method on chat models. We have included the results and corresponding discussions in Section F of the revised manuscript.
>
> Table VI. Performance comparisons between QLLM and OmniQuant for chat models.
> | Model            |        Case        | Former Win |  Tie  | Former Lost |
> | ---------------- | :----------------: | :--------: | :---: | :---------: |
> | LLaMA-2-7B-Chat  | QLLM vs. OmniQuant |  **137**   |  19   |      4      |
> | LLaMA-2-13B-Chat | QLLM vs. OmniQuant |  **116**   |  24   |     20      |
>
>
> **Reference**
>
> [A] Vicuna: An open-source chatbot impressing gpt-4 with 90%* chatgpt quality, March 2023. https: //lmsys.org/blog/2023-03-30-vicuna/.
>
> [B] Judging llm-as-a-judge with mt-bench and chatbot arena. arXiv 2023.
>
> **Q6**. Sec 4.1.1 uses $\mathrm{max}(x_{M})$ to determine the number of disassembly channels $T$ according to $\theta$, why don't we use $\mathrm{max}(| x_{M} |)$?
>
> **A6**. We appreciate your attention. To clarify, we indeed utilize $\mathrm{max}(| x_{M} |)$ to determine the number of disassembly channels $T$ based on $\theta$. We have updated our manuscript accordingly.

---

> ### Author Response · Authors · 2023-11-19
> **Response to Reviewer KacJ (Part 2)**
>
> **Q7**. I wonder whether the CD results in Table 2 use efficient error correction (EEC)? I would like to see the results of using channel disassembly without EEC as a reference.
>
> **A7**. The channel disassembly (CD) results presented in Table 2 indeed incorporate our EEC. To further demonstrate the effectiveness of CD, we apply CD without EEC to obtain 4-bit LLaMA-1-13B and show the results in Table VII. We observe that the absence of both CD and EEC leads to a significant decline in the performance of the quantized model. Notably, using CD alone substantially reduces the performance degradation associated with quantization. Moreover, increasing the channel expansion ratio $\gamma$ further improves the model's performance, which strongly shows the benefits of using CD to decompose the outlier channels. By incorporating both CD and EEC, the performance improvement is even more pronounced, underscoring the efficacy of EEC in conjunction with CD. We have included the results and discussions in Section H of the supplementary material.
>
> Table VII. Perplexity results of channel disassembly (CD) with and without efficient error correction (EEC). “$\gamma$” is the channel expansion ratio. We report the perplexity of W4A4 LLaMA-1-13B on WikiText2, PTB and C4.
> | CD  |  EEC  | $\gamma$ | WikiText2 |    PTB    |    C4     |   Avg.    |
> | --- | :---: | :------: | :-------: | :-------: | :-------: | :-------: |
> |     |       |    -     |  1702.34  |  1853.58  |  1159.41  |  1571.78  |
> | ✓   |       |   0.01   |   19.34   |   45.36   |   23.25   |   29.32   |
> | ✓   |   ✓   |   0.01   | **8.31**  | **14.44** | **10.74** | **11.16** |
> | ✓   |       |   0.03   |   12.11   |   24.73   |   14.38   |   17.07   |
> | ✓   |   ✓   |   0.03   | **8.01**  | **13.52** | **10.27** | **10.60** |
> | ✓   |       |   0.05   |   11.4    |   23.53   |   13.62   |   16.18   |
> | ✓   |   ✓   |   0.05   | **7.85**  | **13.38** | **10.13** | **10.45** |
> | ✓   |       |   0.07   |   11.13   |   23.47   |   13.45   |   16.02   |
> | ✓   |   ✓   |   0.07   | **7.81**  | **13.35** | **10.11** | **10.42** |

---

> ### Author Response · Authors · 2023-11-21
> **Follow-up on Rebuttal**
>
> Dear Reviewer KacJ
>
> We sincerely appreciate your considerable efforts in reviewing our paper. We have provided responses to your concerns, particularly regarding the inference efficiency of QLLM and morel results on the efficient error correction only. If there are any further concerns or questions, please do not hesitate to let us know.
>
> Best regards,
>
> Authors of #54

---

> > ### Comment · Reviewer_KacJ · 2023-11-21
> > **Thanks for the detailed response**
> >
> > Thanks for the detailed response and the added experiment. It addresses most of my concerns. I'm raising the score to 6.

---

> > > ### Author Response · Authors · 2023-11-22
> > > **Thanks for your feedback**
> > >
> > > Dear Reviewer KacJ,
> > >
> > > Thanks for your feedback! We are pleased to address your concerns. Your reviews have been instrumental in improving the quality of our work.
> > >
> > > Best regards,
> > >
> > > Authors of Paper #54

---

### Official Review · Reviewer_dSZP · 2023-10-31

**Soundness:** 3 good
**Presentation:** 2 fair
**Contribution:** 3 good
**Rating:** 6
**Confidence:** 5

**Summary:**

The paper is trying to address the challenge of LLM quantization associated with the well-known "channel-specific outlier problem", i.e. some specific channels in activations tend to have much larger range compared to the others, which will cause difficulty in choosing quantization scaling factor and degradation in accuracy. The main solution proposed is to disassemble an outlier channel into T channels, therefore, the magnitude of each channel becomes 1/T of the original channel. In order not to increase the computational cost by too much, the second part of the proposed method is to search for similar channels and merge them by averaging the activations and summation of the corresponding weights. As a result, total number of channels will be the same as the original tensors after these channel disassembly and assembly steps. Furthermore, a LORA/QLORA-like tuning is applied to greatly alleviate the cost of PTQ while enabling simultaneous tuning of multiple blocks, which is critical to improve model performance (perplexity). Finally, the inference benchmark results show that the proposed method will add ~20% overhead compared to INT8.

**Strengths:**

1. The proposed channel decomposition/disassembly method is simple and effective.
2. The method to determine disassembly threshold (theta) is reasonable and computationally affordable.
3. Outperforms previous methods at W4A4 on Llama-1 and Llama-2.
4. acceptable overhead compared to INT8.

**Weaknesses:**

1. W6A6 results are included but only referenced a few times, mostly used in a sentence like "QLLM is comparable with other methods at W6A6." Given that authors in general would like to impress readers by "being better than others" rather than "being comparable to others," W6A6 really doesn't make a strong point in the main article. Plus W6A6 has almost no real HW support... The author might want to consider adding a few more comments/discussion on W6A6 results or moving them to appendix and make the main article more concise.
2. A little clarification about the inference time benchmark would be helpful. For example, readers might be interested in comparing the QLLM results with other quantization implementations, like weight-only quantization W4A16. Take auto-gptq (https://github.com/PanQiWei/AutoGPTQ/tree/main#inference-speed) as a reference, the "speed" usually is in the unit of token/sec or msec/token. Table 4 only says "inference time" and the unit is ms, which is a little unclear.

**Questions:**

1. Table 2 shows "Channel Disassembly only" approach with just 1% of channel expansion ratio can already achieve comparable results with the final method. In fact, the reason to use channel assembly is mainly to reduce the computational cost. Author may want to add some comments or examples regarding the overhead incurred by this 1% extra channels, in order to justify the need for Channel Assembly method.
2. The paragraph of "Inference efficiency" as well as Table 4 didn't specify whether this "FP" model is FP32 or FP16. If it's FP32, author may also want to include FP16 results since FP16 is used as the baseline on accuracy tables. Also it is understandable that the implementation for INT computation may not be optimized. Therefore, instead of comparing the absolute run time, it would be helpful to include the additional Ops of QLLM compared to INT8.

---

> ### Author Response · Authors · 2023-11-19
> **Response to Reviewer dSZP**
>
> Thanks for your constructive comments.
>
> **Q1**. W6A6 really doesn't make a strong point in the main article. Plus W6A6 has almost no real HW support. The author might want to consider adding a few more comments/discussions on W6A6 results or moving them to the appendix and making the main article more concise.
>
> **A1**. Thanks for your advice. Following your suggestion, we expanded our discussions on W6A6 results. We acknowledge that W6A6 has limited hardware support in real-world applications. However, our QLLM still demonstrates performance benefits in these settings. For example, QLLM consistently surpasses the state-of-the-art quantization method in terms of lower perplexity across all models on both WikiText2 and C4 and achieves comparable accuracy on 5 zero-shot tasks, as shown in Tables 1 and A. While the absolute performance gains with 6-bit quantization might seem modest, this is partly due to the less pronounced effect of activation outliers at this bitwidth. Our approach's benefits are more pronounced in lower bitwidth scenarios, especially under W4A4 scenarios, as demonstrated in Tables 1, 5, and A. We have included the discussions in Section 5.1 of the revision.
>
> **Q2**. A little clarification about the inference time benchmark would be helpful. For example, readers might be interested in comparing the QLLM results with other quantization implementations, like weight-only quantization W4A16.
>
> **A2**. Following your suggestion, we compare our QLLM with weight-only quantization W4A16. Please refer to Q1 of the general response for the inference time results. Notably, even when compared to the already efficient W4A16, our QLLM still achieves a **1.1$\times$ speedup**, underscoring its superior efficiency.
>
> **Q3.** Author may want to add some comments or examples regarding the overhead incurred by 1% extra channels from channel disassembly, in order to justify the need for channel assembly method.
>
> **A3**. Please refer to Q1 in the general response.
>
> **Q4**. The paragraph of "Inference efficiency" as well as Table 4 didn't specify whether this "FP" model is FP32 or FP16.
>
> **A4**. The FP in Table 4 denotes the FP16 model. We have revised our manuscript.
>
> **Q5**. It would be helpful to include the additional Ops of QLLM compared to INT8.
>
> **A5**. Following your suggestion, we measure the computational complexity of the quantized models using the Bit-Operation (BOP) count [A]. From Table IV, our 8-bit QLLM introduces only a slight increase in BOPs compared to the INT8 model but substantially lower than those of the FP16 counterpart. We have included the results and discussions in Section M of the supplementary material.
>
> Table IV. BOPs comparisons of different models. We report the results of LLaMA-1-7B with a mini-batch size of 1.
> | $L$  |      256      |      512      |     1024      |      2048      |
> | ---- | :-----------: | :-----------: | :-----------: | :------------: |
> | FP16 |    875.52T    |   1,766.40T   |   3,604.48T   |   7,493.12T    |
> | INT8 |    231.58T    |    467.64T    |    952.56T    |    1976.16T    |
> | QLLM | 231.58T+1.07M | 467.64T+2.14M | 952.56T+4.28M | 1976.16T+8.56M |
>
> **Reference**
>
> [A] Differentiable Joint Pruning and Quantization for Hardware Efficiency. ECCV 2020.

---

> > ### Comment · Reviewer_dSZP · 2023-11-21
> >
> > My rating remains the same for this paper. Thanks for all the explanations, responses, and new results!

---

> > > ### Author Response · Authors · 2023-11-22
> > > **Thanks for your feedback**
> > >
> > > Dear Reviewer dSZP,
> > >
> > > Thank you for your feedback! We appreciate the constructive reviews for improving our work.
> > >
> > > Best regards,
> > >
> > > Authors of Paper #54

---

### Author Response · Authors · 2023-11-19
**Response to All Reviewers (Part 1)**

We sincerely thank all reviewers for their valuable comments.

## Reasonable and Effective Method
All reviewers agree that

* “*The proposed channel decomposition/disassembly method is simple and effective. The method to determine disassembly threshold (theta) is reasonable and computationally affordable*”  (Reviewer dSZP)
* ”*Channel splitting is a reasonable method to address the activation outlier issue.*” (Reviewer KacJ)
* “ *… provide evidence for the effectiveness of CD and CA.*” (Reviewer wUVT)
* “*... showing the effectiveness of each component. The efficient fine-tuning method seems like an efficient way to restore performance.*” (Reviewer rXfg)

## General Response
**Q1**. Inference overhead of channel disassembly and assembly. (Reviewers dSZP, KacJ, wUVT, rXfg)

**A1**. The inference overhead associated with channel disassembly and assembly in our model is small for several reasons:
1) As mentioned in Section 1, recent studies (Dettmers et al., 2022; Xiao et al., 2023; Wei et al., 2023) have revealed that activation outliers are often concentrated in specific channels across various inputs. Leveraging this insight, we are able to pre-calculate the channel indices for disassembly and assembly using a small number of calibration data, which significantly reduces runtime overhead.
2) As mentioned in Section 4.3, our channel disassembly and assembly can be implemented efficiently. If the previous layer is a linear layer, both channel disassembly and assembly can be seamlessly fused into the linear layer, which eliminates additional inference overhead. In cases where the preceding layer is a non-linear layer, additional disassembly and assembly layers are inserted to decompose and aggregate channels during runtime. Benefiting from our efficient kernel implemented by Triton [A] and the limited reassembly ratio searched by our adaptive strategy (See Figure A in the initial submission), the introduced inference cost is controlled within a small level, i.e., **4% additional cost**.

To demonstrate this, we employ W4A4 kernels from [QUIK](https://github.com/IST-DASLab/QUIK) codebase and reassess the inference efficiency. We also conduct a comparative analysis using weight quantization only, utilizing CUDA kernels from [AutoGPTQ](https://github.com/PanQiWei/AutoGPTQ). As shown in Table I, channel disassembly results in additional inference costs due to the extra channels. These additional channels often don't align with GPU-friendly multiples like 32 or 64, leading to less efficient GPU use. Using our channel assembly method maintains the original channel count, ensuring better GPU utilization and mitigating the extra inference costs from disassembly. As a result, the quantized models with both channel disassembly and assembly achieve higher throughput compared to the ones with disassembly only, which demonstrates the necessity of channel assembly. Importantly, our 4-bit QLLM incurs only **4% additional cost** relative to W4A4 fixed-point baseline, while achieving a notable **1.96$\times$ speedup** over FP16. We have included the inference results and discussions in Section 5.2 and Section M of the supplementary material. We plan to release our efficient Triton kernel to the community to facilitate further research and development in this area.

Table I. Inference throughput (tokens/s) comparisons of different models. The throughput is measured with a 2048-token segment on NVIDIA RTX 3090 GPUs: 1x GPU for LLaMA-1-7B and 2x GPUs for LLaMA-1-13B. “CD” stands for channel disassembly. “CA” represents channel assembly. “Adaptive” refers to the adaptive strategy. “$\gamma$” is the channel expansion ratio.

| Model | Method | CD | CA | Adaptive | $\gamma$ | Inference Throughput (tokens/s) |
| - | - | :-: | :-: | :-: | :-: | :-: |
| LLaMA-1-7B | FP16 | |  |  | –  | 3252 |
| LLaMA-1-7B | W8A8 | | |  | – | 5676 |
| LLaMA-1-7B | W4A16 |  |  |  | – | 5708 |
| LLaMA-1-7B | W4A4 |  |   |   | – | 6667 |
| LLaMA-1-7B | W4A4 |✓|  | |0.01| 6322 |
| LLaMA-1-7B | W4A4 |✓| |  |0.05| 6315 |
| LLaMA-1-7B | W4A4 |✓|  |  | 0.1 | 6310 |
| LLaMA-1-7B | W4A4 |✓| ✓|  |0.01| 6365 |
| LLaMA-1-7B  | W4A4|✓| ✓|  |0.05| 6334|
| LLaMA-1-7B  | W4A4|✓|✓| |0.1|6318|
| LLaMA-1-7B  | W4A4| ✓ | ✓ |✓| - |6385|
| LLaMA-1-13B | FP16|  |  |  |–  |1910|
| LLaMA-1-13B | W8A8 | |    |  |  –  |3179|
| LLaMA-1-13B | W4A16|  |   |  | –  |2026|
| LLaMA-1-13B | W4A4| |  |  | –  |3873|
| LLaMA-1-13B | W4A4|   ✓   |  |    |   0.01   |3728 |
| LLaMA-1-13B | W4A4|   ✓   |  |   |   0.05   |3725 |
| LLaMA-1-13B | W4A4|   ✓   |  |    |   0.1    |3678 |
| LLaMA-1-13B | W4A4|   ✓   |   ✓   |          |   0.01   |3731 |
| LLaMA-1-13B | W4A4|   ✓   |   ✓   |          |   0.05   | 3728|
| LLaMA-1-13B | W4A4|   ✓   |   ✓   |          |   0.1    | 3681 |
| LLaMA-1-13B | W4A4|   ✓   |   ✓   |    ✓     |    --    | 3730 |

**Reference**

[A] Triton: an intermediate language and compiler for tiled neural network computations. MAPL 2019.

---

> ### Author Response · Authors · 2023-11-19
> **Response to All Reviewers (Part 2)**
>
> **Q2**. More results regarding efficient error correction (EEC) without channel reassembly under different numbers of calibration samples. (Reviewers KacJ, wUVT)
>
> **A2**. Using EEC only without our channel reassembly results in suboptimal performance as it suffers from activation outlier issues. To demonstrate this, we applied EEC only to quantize LLaMA-1-7B to 4-bit, using the same training settings as our QLLM but with varying numbers of calibration samples. From Table II, even with an increased amount of calibration data, the performance of the EEC only significantly lags behind our QLLM. These results strongly demonstrate the effectiveness of channel reassembly in addressing activation outliers, thereby substantially improving performance. We have included the results and corresponding discussions in Section I of the supplementary material.
>
> Table II. Performance comparisons with different methods under various numbers of calibration samples. We report the perplexity of W4A4 LLaMA-1-7B on WikiText2, PTB and C4.
>
> | Method | #Samples | WikiText2 |    PTB    |    C4     |   Avg.    |
> | ------ | :------: | :-------: | :-------: | :-------: | :-------: |
> | EEC    |   128    |   16.64   |   38.58   |   28.33   |   27.85   |
> | EEC    |   256    |   14.94   |   37.70   |   33.62   |   28.75   |
> | EEC    |   512    |   12.35   |   29.39   |   31.59   |   24.44   |
> | QLLM   |   128    | **9.65**  | **16.56** | **12.29** | **12.83** |
>
>
> **Q3**. The advantage of EEC over tuning quantized weights directly (TQW). (Reviewers wUVT, rXfg)
>
> **A3**. As mentioned in Section 4.2, compared with TQW which tunes all quantized weights, our EEC focuses on learning a small set of low-rank weights, which significantly reduces training costs and GPU memory usage while delivering comparable performance. The efficiency advantages of EEC are particularly notable in the context of larger models. For example, when applying EEC and TQW to the 4-bit LLaMA-1-65B model, as shown in Table III, EEC with even better perplexity demonstrates a **1.56$\times$ faster training time** and a **1.62$\times$ reduction in GPU memory usage** compared to TQW for 1 Attn-FFN Block reconstruction. Moreover, the reduced GPU memory demand allows EEC to quantize LLaMA-1-65B on a single 24GB consumer-grade GPU, such as the NVIDIA RTX 4090, a task that is not feasible with TQW. We have included these results and the corresponding discussions in Section 5.2.
>
> Table III. Perplexity, training time, and GPU memory comparisons between efficient error correction (EEC) and tuning quantized weights directly (TQW) for 4-bit LLaMA-1-65B. “OOM” indicates out of memory.
>
> | #Attn-FFN Block | Method | WikiText2 |  PTB  |  C4   | Avg.  | Training Time (GPU Hours) | GPU Memory (GB) |
> | :-------------: | ------ | :-------: | :---: | :---: | :---: | :-----------------------: | :-------------: |
> |        1        | TQW    |   6.34    | 17.61 | 9.56  | 11.17 |           12.16           |      30.84      |
> |        1        | EEC    |   8.31    | 13.77 | 10.76 | 10.95 |         **7.79**          |    **19.00**    |
> |        2        | TQW    |   6.25    | 11.18 | 8.56  | 8.66  |           12.13           |      52.45      |
> |        2        | EEC    |   7.62    | 11.47 | 9.39  | 9.49  |         **7.79**          |    **28.60**    |
> |        4        | TQW    |     -     |   -   |   -   |   -   |            OOM            |        -        |
> |        4        | EEC    |   6.87    | 11.36 | 8.98  | 9.07  |         **7.77**          |    **47.71**    |
>
> ## Summary of changes
> We have revised our submission and summarized our updates as follows:
> * We have implemented an efficient Triton kernel for our channel reassembly and provided more inference overhead results of QLLM. (Reviewers dSZP, KacJ, wUVT, and rXfg)
> * We have conducted more empirical studies on 1) the effect of the efficient error correction without channel reassembly (Reviewers KacJ and wUVT); 2) the effect of the channel disassembly without the efficient error correction (Reviewer KacJ); 3) the block-wise reconstruction error before and after applying channel reassembly (Reviewer KacJ); 4) the effect of the weight merging in the efficient error correction
> * We have discussed the benefit of efficient error correction over tuning quantized weights directly. (Reviewers wUVT and rXfg)
> * We have provided more discussion on W6A6 results. (Reviewers dSZP and rXfg)
> * We have conducted more experiments on chat models. (Reviewer KacJ)
> * We have provided more details on the pseudo-codes of channel disassembly and assembly during runtime. (Reviewer wUVT)

---

> ### Comment · Reviewer_wUVT · 2023-11-21
> **Re: Response to All Reviewers (Part 2)**
>
> I would like to express my gratitude to the authors for providing detailed feedback. However, I still have some concerns regarding the effectiveness of the proposed method.
> - Firstly, the reported results of QLLM in Table II of **A2** appear to be unconvincing. It is noteworthy that the perplexity of W4A4 LLaMA-1-7B on WikiText2, PTB, and C4 with QLLM is exactly the same as the results of TQW with 4 attn-FFN block tuning, as reported in Table 6 of the original submission. Is this merely a coincidence?
>
> - Secondly, I agree with the authors' statement that "the efficiency advantages of EEC are particularly notable in the context of larger models." However, the authors have not mentioned that the performance drop between TQW and the proposed EEC also becomes notable in the context of large models. Additionally, it seems that the TQW 4-bit model exhibits faster runtime speed compared to QLLM. Furthermore, the training setting of TQW is much simpler than the proposed two-step tuning. Considering that the quantization process is commonly deployed on cloud servers, I personally believe that the 12.13 GPU hours required for TQW are entirely acceptable.
>
> - Lastly, I would like to inquire whether the "LLaMA-1-7B W4A16" model still suffers from the outlier issue. In my opinion, the A4/A16 mix-precision setting could potentially serve as a strong baseline method.

---

> > ### Author Response · Authors · 2023-11-22
> > **Response to additional questions (Part 1)**
> >
> > Thank you for your feedback.
> >
> > **Q4**. The reported results of QLLM in Table II of A2 appear to be unconvincing.
> >
> > **A4**. Thanks for pointing out. The PPL of QLLM on WikiText2, PTB, C4 should be 9.65, 16.56, and 12.29, respectively. We have updated the responses and revision accordingly.
> >
> > **Q5**. I agree with the authors' statement that "the efficiency advantages of EEC are particularly notable in the context of larger models." However, the authors have not mentioned that the performance drop between TQW and the proposed EEC also becomes notable in the context of large models. Additionally, it seems that the TQW 4-bit model exhibits faster runtime speed compared to QLLM. Furthermore, the training setting of TQW is much simpler than the proposed two-step tuning. Considering that the quantization process is commonly deployed on cloud servers, I personally believe that the 12.13 GPU hours required for TQW are entirely acceptable.
> >
> > **A5**. We would like to highlight that both TQW and EEC **follow the same two-step strategy, which initially applies channel reassembly followed by a reconstruction phase to improve the performance of quantized LLMs**. As a result, the runtime speed for TQW and QLLM are **identical** because they both apply the same **post-reassembly quantization model**. It is important to note that the effectiveness of TQW is highly dependent on our channel reassembly. Without channel reassembly, TQW would exhibit significant performance degradation due to activation outlier issue, as shown in Table X below.
> >
> > Compared with TQW that tunes all quantized weights, our EEC is specifically designed for memory efficiency, focusing on learning a small set of low-rank weights. This efficiency is especially valuable in cloud computing settings where access to high-end GPUs is not available. EEC’s efficient design enables the quantization of large models, such as LLaMA-1-65B, on a single 24GB GPU like the NVIDIA A30 or A10, which is not feasible with TQW. This makes EEC an invaluable tool for communities with limited resources, significantly broadening the accessibility and practicality of quantization techniques for LLMs.
> >
> > Table X. Performance comparisons with different methods. We report the perplexity of 4-bit LLaMA-1-7B on WikiText2, PTB and C4. ‘’CR’’ denotes our adaptive channel reassembly.
> > | Method | WikiText2 |  PTB  |  C4   | Avg.  |
> > | ------ | :-------: | :---: | :---: | :---: |
> > |  TQW w/o CR  |  13.13 | 42.81 | 32.07 | 29.34 |
> > |   TQW w/ CR   | **8.90** | **14.75** | **11.63** | **11.76** |
> >
> > **Q6**. I would like to inquire whether the "LLaMA-1-7B W4A16" model still suffers from the outlier issue. In my opinion, the A4/A16 mix-precision setting could potentially serve as a strong baseline method.
> >
> > **A6**. Activation outliers exist irrespective of bitwidths, but their impact on model performance is negligible in the W4A16 setting. We agree that a mixed-precision approach, such as A4/A16, could be a strong baseline. However, mixed-precision within layers incurs additional computational load due to the simultaneous handling of multiple bitwidth multiplications, which can increase noticeable inference overhead compared to a fixed-precision approach for small models, as shown in Table 5 of gpt3.int8 (Dettmers et al., 2022). In contrast, our QLLM adopts a uniform bitwidth, which only introduces little additional overhead compared to the 4-bit fixed-precision baseline (See Table I in the general response).

---

> ### Comment · Reviewer_wUVT · 2023-11-21
> **Re: Response to All Reviewers (Part 1)**
>
> I kindly request the authors to provide additional information regarding the corresponding $\gamma$ ratio in the adaptive strategy. Considering the observed runtime speed, it appears that the actual $\gamma$ ratio in this particular setting is exceedingly small. I am curious to know whether the replicated channels in the adaptive strategy are included in the fixed $\gamma$ settings.

---

> > ### Author Response · Authors · 2023-11-22
> > **Response to additional questions (Part 2)**
> >
> > **Q7**. I kindly request the authors to provide additional information regarding the corresponding $\gamma$ ratio in the adaptive strategy. I am curious to know whether the replicated channels in the adaptive strategy are included in the fixed $\gamma$ settings.
> >
> > **A7**. We have presented the detailed channel expansion ratios $\gamma$ for the input activations of different layers of 4-bit LLaMA-1-13B in Figure A of our initial submission. Additionally, we further provide the searched expansion ratios for 4-bit LLaMA-1-7B in Figure B in the supplementary material of the revision. From the results, $\gamma$ typically falls within the range of [0, 0.1]. For the inference overhead associated with specific $\gamma$ values within this range, we have put it in Table I of the general response.

---

### Meta-Review · Area_Chair_h2Ah · 2023-12-09

**Metareview:**

This paper proposed to address the issue outlier channels in quantizing the language models. The main idea is to disassemble the channels with larger outliers into several segments so that they become easier to quantize. And during run time, the channels are re-assembled together for inference. The idea is simple and seems to be effective with relatively small overhead. With an implemented Triton kernel, the authors showed that it is about 4% overhead, which could be a good tradeoff in different scenarios. The authors did a great job addressing reviewers's concerns in the paper.

Strengths:
1. The idea is simple and seems to be effective in the demonstrated cases.
2. The implemented Triton kernel could be useful for the community.
Weekness:
2. The method is not particular clean, so I doubt the proposed method would stand the test of time or be applied to other models.

**Justification For Why Not Higher Score:**

The novelty of the paper is likely to be small due to its limited scope.

**Justification For Why Not Lower Score:**

All reviewers agreed that this paper could be a good addition to the conference.

---

### Decision · Program_Chairs · 2024-01-16

Accept (poster)